# 3-O Sulfated Heparan Sulfate (G2) Peptide Ligand Impairs the Infectivity of *Chlamydia muridarum*

**DOI:** 10.3390/biom15070999

**Published:** 2025-07-12

**Authors:** Weronika Hanusiak, Purva Khodke, Jocelyn Mayen, Kennedy Van, Ira Sigar, Balbina J. Plotkin, Amber Kaminski, James Elste, Bajarang Vasant Kumbhar, Vaibhav Tiwari

**Affiliations:** 1Arizona College of Osteopathic Medicine, Midwestern University, Glendale, AZ 85308, USA; weronika.hanusiak@midwestern.edu; 2Department of Biological Sciences, Sunandan Divatia School of Science, SVKM’s Narsee Monjee Institute of Management Studies (NMIMS) Deemed-to-Be University, Mumbai 400056, Maharashtra, India; khodkepurva@gmail.com; 3Department of Microbiology & Immunology, Midwestern University, Downers Grove, IL 60515, USA; jocelyn.mayen@midwestern.edu (J.M.); kennedy.van@midwestern.edu (K.V.); bplotk@midwestern.edu (B.J.P.);

**Keywords:** *Chlamydia*, infectivity, intracellular, computational modeling, molecular dynamics

## Abstract

**Background:** Heparan sulfate (HS) is widely implicated as a receptor for *Chlamydia* cell attachment and infectivity. However, the enzymatic modification of HS modified by the 3-O sulfotransferase-3 (3-OST-3) enzyme in chlamydial cell entry remains unknown. **Methodology:** To rule out the possibility that host cell 3-O sulfated heparan sulfate (3-OS HS) plays a significant role in *C. muridarum* entry, a Chinese hamster ovary (CHO-K1) cell model lacking endogenous 3-OST-3 was used. In addition, we further tested the efficacy of the phage-display-derived cationic peptides recognizing heparan sulfate (G1 peptide) and the moieties of 3-O sulfated heparan sulfate (G2 peptide) against *C. muridarum* entry using human cervical adenocarcinoma (HeLa 229) and human vaginal epithelial (VK2/E6E7) cell lines. Furthermore, molecular dynamics simulations were conducted to investigate the interactions of the *Chlamydia* lipid bilayer membrane with the G1 and G2 peptides, focusing on their binding modes and affinities. **Results:** The converse effect of 3-OST-3 expression in the CHO-K1 cells had no enhancing effect on *C. muridarum* entry. The G2 peptide significantly (>80%) affected the cell infectivity of the elementary bodies (EBs) at all the tested concentrations, as evident from the reduced fluorescent staining in the number of inclusion bodies. The observed neutralization effect of G2 peptide on *C. muridarum* entry suggests the possibility of sulfated-like domains being present on the EBs. In addition, data generated from our in silico computational structural modeling indicated that the G2 peptide ligand had significant affinity towards the *C. muridarum* lipid bilayer. **Conclusions:** Taken together, our findings show that the pretreatment of *C. muridarum* with 3-O sulfated heparan sulfate recognizing G2 peptide significantly prevents the entry of EBs into host cells.

## 1. Introduction

*Chlamydia* spp. is a gram-negative, obligate intracellular mucosal pathogen, with a broad host and organ-site infection range, responsible for a wide range of diseases [1,2]. While *C. trachomatis* serovars A, B, Ba and C are the leading causes of infectious blindness in the non-industrial world (keratoconjunctivitis), *C. trachomatis* serovars L1–L3 and D–K cause urogenital infections, including lymphogranuloma venereum (LGV), cervicitis, urethritis, including serious sequelae in the form of pelvic inflammatory disease (PID), and salpingitis, resulting in infertility and ectopic pregnancy [3]. Additionally, exposure to *C. pneumoniae* can lead to the development of acute upper and lower respiratory illnesses, including bronchitis, pharyngitis, sinusitis and pneumonia [4]. The murine-adapted strain, *C. muridarum,* is commonly used as a rodent model for studying human female genital tract infections caused by *Chlamydia*, and mirrors observations made concerning the upper genital tract pathology of *C. trachomatis* in human female genital infections [5].

Regardless of the host, the infectious cycle of *Chlamydia* begins in the epithelium of the targeted tissues with the attachment and internalization of infectious elementary bodies (EBs) [1]. During the initial attachment process, *Chlamydia* spp. uses host cell surface heparan sulfate (HS) [6,7,8]. In addition, *Chlamydia* infection is enhanced in a heparan sulfate proteoglycan (HSPG)-dependent manner via the fibroblast growth factor 2 (FGF2)-mediated signaling pathway [9]. Furthermore, the level and position of sulfation in HSPGs appear to play a role in the binding of *C. muridarum* and *C. trachomatis* L2 to epithelial cells [10].

Structurally, HS is composed of a core protein covalently linked to the chains of unbranched sulfated anionic polysaccharides [11]. Modification of the HS chain is vital in mediating critical interactions with multiple ligands [12]. One such modification is the generation of unique sulfate moieties, such as N-/2-O/6-O- and 3-O-sulfated HS, which play a role in maintaining cellular homeostasis [13]. The final step in this HS modification is carried out by 3-O sulfotransferase (3-OST-3) at the C3 position of the glucosamine residue [14]. These modifications are essential in multiple biological processes as modified HS provides specific binding sites for a variety of viral and bacterial proteins, including adhesion molecules [15,16]. For example, *Chlamydia* hijacks host HS receptors during cell entry [6,7,8]. Thus, HS and their enzymatically sulfated variants provide a molecular basis for potentially targeting anti-chlamydial therapy [17,18,19].

Previous studies have shown that the initial interaction between EBs and the target cell can be significantly blocked by the addition of excess soluble HS, recombinant cysteine-rich outer membrane complex protein (OmcB), or anti-OmcB antibody [20,21]. Furthermore, the removal of 6-O sulfation by 6-O-endosulfatases significantly decreases *C. muridarum* EB attachment and vacuole formation [22]. However, the role of 3-O sulfation in heparan sulfate in *C. muridarum* infectivity has not been determined. The goal of the current study was to determine whether 3-O sulfation in heparan sulfate also plays a role in *C. muridarum* attachment and entry. In this direction, we first tested whether the cell surface expression of 3-O sulfated heparan sulfate (3-OS HS) would have any impact on the infectivity of *C. muridarum*. Our data clearly showed that the presence of the 3-OS HS receptor did not enhance the infectivity of *C. muridarum*. Since HS contains many negatively charged sulfate groups, it was logical to test whether HS- and 3-OS HS-recognizing G1/G2 peptides would influence the entry of Chlamydia. The G1 and G2 peptides were originally isolated to generate glycomimetic reagents that would bind to cellular heparan sulfate (G1 peptide) and its 3-O-sulfated heparan sulfate (G2 peptide) site [23]. One additional reason to test these peptides against Chlamydia was the fact that the above two cationic peptides were previously shown to inhibit the entry of herpes simplex virus type 1 (HSV-1) and Cytomegalovirus (CMV) [23]. The above pathogens, along with *Chlamydia*, are recognized as important agents in venereal and neonatal diseases [24,25,26]. It is also worth noting that the G1/G2 peptide provides a unique tool for studying multiple other pathologies, such as Alzheimer’s disease, hemorrhagic shock, and prostate, breast and hepatocellular cancer, in which enhanced expression of 3-O sulfation has been reported [27,28,29,30,31].

Structurally, the G1 peptide (LRSRTKIIRIRH) and the G2 peptide (MPRRRRIRRRQK) used in this study were distinct. For example, G1 peptide possesses alternating positively charged residues while G2 peptide has strings of cationic residues [23]. Interestingly, our previous findings have shown that G2 peptide preferentially recognizes sulfated domains on the HS chain [32]. In this study, we took two independent experimental approaches. In the first approach, EBs of *C. muridarum* were pre-exposed before being challenged with human-derived cervical and/or vaginal epithelial cell lines. In the second approach, the target cells were first exposed to G1 and/or G2 peptides before they were challenged with *C. muridarum.* Our results found that the application of the pre-treatment with G2 peptide to Chlamydia’s EBs showed a robust inhibition in the entry of *C. muridarum* as compared to G1 peptide in both the tested cell lines. On the other hand, the pre-treatment of target cells with G1/G2 peptides was not as effective in terms of cell infectivity compared to pre-treatment of G1 or G2 peptide to *Chlamydia’s* EBs. The above data were further supported by molecular dynamics simulation studies, which showed that the G2 peptide interacted better with the lipid bilayer membrane of *C. muridarum* than the G1 peptide.

## 2. Materials and Methods

### 2.1. Cell Cultures

The three cell lines used in this study were purchased from ATCC (American Type Culture Collection). Two were human female genital tissue-derived cell lines, i.e., HeLa 229 (human epithelial cervix adenocarcinoma, ATCC CCL-2.1^TM^) and VK2/E6E7 (human vaginal epithelial; ATCC CRL-2616™) cell lines. The third cell line, i.e., Chinese hamster ovary (CHO-K1; ATCC CCL-61™), served as the control to investigate the role of the 3-O-sulfated heparan sulfate (3-OS HS), as it lacks the 3-OST-3 enzyme responsible for generating 3-OS HS receptor [33]. HeLa 229 cells were grown at 37 °C with 5% CO_2_ in complete Dulbecco’s Modified Eagle Medium (DMEM; Corning Cell-gro) with 4.5 g/L glucose, 10% fetal bovine serum (VWR International), 110 mg/L L-glutamine and 1% gentamicin (HyClone). VK2/E6E7 cells were grown in Keratinocyte serum-free media (Life Technologies) supplemented with 0.1 ng/mL human recombinant epidermal growth factor (EGF), 0.05 mg/mL bovine pituitary extract, 0.4 mM calcium chloride and penicillin/streptomycin (Gibco/BRL).

### 2.2. Expression of 3-O Sulfotransferase-3 (3-OST-3)

To determine whether host cell surface 3-O-sulfated heparan sulfate (3-OS HS) plays a role in *C. muridarum* entry, a CHO-K1 cell model lacking endogenous 3-O-sulfotransferase-3 (3-OST-3) was used. Wild-type CHO-K1 cells were grown in Ham’s F-12 medium (Gibco/BRL, Carlsbad, CA, USA) supplemented with 10% fetal bovine serum (FBS), penicillin and streptomycin (Gibco/BRL). CHO-K1 cells were transfected with human encoded 3-OST-3 mammalian expression plasmid (OriGene) at 1.5 µg/µL using lipofectamine 2000 (Invitrogen), according to the manufacturer’s recommendations. CHO-K1 cells transfected in parallel with an empty vector (pcDNA3.1) plasmid were used as a negative control. After 72 h, post-transfected cells were challenged with *C. muridarum* to compare the cell infectivity in the presence and absence of 3-OST-3 enzyme.

### 2.3. C. muridarum Infectivity Assay

For infectivity assays, *C. muridarum* Nigg strain from a frozen −80 °C stock at multiplicity of infection (MOI) of 1.0 was used. *C. muridarum* Nigg strain was originally obtained from Dr Roger Rank of the University of Arkansas, who obtained it from American Type Culture Collection (Rockville, MD, USA). Following 24 h incubation, cells were fixed with methanol and stained with mouse anti-*C. muridarum* primary antibody and goat anti-mouse IgG secondary antibody (Invitrogen). Images of *C. muridarum* inclusions were captured at 10× using an EVOS FL Auto Imaging fluorescent microscope (Life Technologies). The level of infectivity was determined by counting the number of inclusion-forming units (IFU) in a minimum of 10 high-power fields (Zeiss Axiovert 25 fluorescent microscope) for each replicate with a minimum of three replicates per treatment to calculate the IFU/mL for each sample. The IFU/mL was calculated as follows: total number of inclusions per sample X sample dilution factor X field factor. Field factors vary depending on: (i) the number of fields counted; (ii) the well area in each plate; (iii) the microscope magnification. For these studies, the field factor is 6.15 for a 96-well plate (Genesee Scientific Corporation, El Cajon, CA, USA) with 10 counted fields at a magnification of 40×.

### 2.4. Chemical Structure of G1 and G2 Peptide Ligands

The 12-amino acid residue (12-mer) G1 and G2 peptide ligands were generated against plain-type HS and 3-O sulfated HS and synthesized according to a previously reported scheme [23]. The peptide sequence and characterization/purification of G1 and G2 peptide ligands are shown in Figure 1.

### 2.5. Pretreatment of C. muridarum with G1 and or G2 Peptide Ligands

The cell lines tested were grown in 96-well plates for 24 h to a confluency of 100%. The elementary bodies of *C. muridarum* (Nigg strain) at a MOI of 1.0 were pre-treated with HS (G1) peptide ligand or 3-OS HS (G2) peptide ligand in a two-fold-dependent manner, ranging from 0.00036 µM to 0.0030 µM, followed by rocking for 1 h (Scigene hybridization incubator, 4 RPM at 35 °C). The supernatant was removed by aspiration, and DMEM containing either G1 or G2 peptide ligand-treated *C. muridarum* was added (50 µL) to the target cells. Untreated *C. muridarum* (MOI 1.0, 50 µL) was added to cells as a positive control. Plates containing cells with G1- or G2-treated *C. muridarum* and controls were centrifuged for 1 h (1100× *g*, 35 °C). After centrifugation, the supernatant was removed by aspiration. Fresh, complete DMEM (100 µL) with 0.5 µg/mL cycloheximide was then added, and the plates were incubated for 24 h before fixing and staining the cells for fluorescent imaging of inclusion bodies. To understand the affinity of G1 and G2 peptides with the lipid bilayer of *C. muridarum*, molecular modeling and molecular dynamics simulations were employed.

### 2.6. Target Cell Pretreatment with G1 and or G2 Peptide Ligands

Target HeLa and VK2 cells were cultured in 96-well plates for 24 h to a confluency of 100%. The cells were independently pre-treated with HS (G1) peptide ligand and/or 3-OS HS (G2) peptide ligand in a two-fold-dependent manner ranging from 0.00036 to 0.0030 μM, followed by rocking for 1 h. Supernatant was removed by aspiration. The cells were then challenged with *C. muridarum* (Nigg strain) at a MOI of 1.0 for 24 h before fixing and staining the cells for fluorescent imaging of inclusion bodies, as described above. Untreated cells infected with *C. muridarum* at MOI of 1.0 were considered as a positive controls.

### 2.7. Exploring the Binding Mode and Affinity of G1 and G2 Peptides with the Lipid Bilayer Membrane Complex of C. muridarum Using Molecular Dynamics Simulation

#### 2.7.1. G1 and G2 Peptide Modeling

The three-dimensional structures of the HS (G1-LRSRTKIIRIRH) and 3-O-sulfated HS (G2-MPRRRRIRRRQK) peptide ligands were predicted using the I-TASSER server, which employs a homology-based multiple-threading strategy to generate atomic-level 3D models of peptides [34]. The confidence score, TM-score and RMSD were used to obtain the best-fit model for both the G1 and G2 peptides. Further, molecular dynamics (MD) simulations were conducted to examine the dynamic behavior and stability of the peptides in interaction with the *C. muridarum* membrane.

#### 2.7.2. *C. muridarum* Membrane Generation and MD Simulation

To elucidate the interaction and stability of G1 and G2 peptides with the *C. muridarum* membrane, MD simulations were performed using Gromacs 2021.5 [35]. The CHARMM-GUI interface [36] was used to construct these MD simulation systems, including the G1-membrane and G2-membrane complexes. Both peptides, G1 and G2, were initially positioned 20 Å above the *C. muridarum* membrane surface using translation and flip operations along the Z-axis. These MD simulation systems were solvated in a rectangular box filled with TIP3P water molecules, with K+ and Cl- ions added at a concentration of 0.15 M to maintain charge neutrality and physiological conditions. To model the *C. muridarum* outer membrane, we utilized lipid composition data obtained from the Orientations of Proteins in Membranes (OPM) database [37]. This Gram-negative bacterial pathogen possesses a unique trilaminar outer membrane. The inner leaflet of our model incorporated a mixture of phosphatidylethanolamine (PE), phosphatidylglycerol (PG) and cardiolipin (CL) lipid headgroups in a 75:20:5 ratio. Given that *Chlamydia* species feature a truncated LPS antigen (LPSA) in their outer membrane, characterized by a group-specific epitope composed of a KDO trisaccharide (K-KDO-(2C8)-K-KDO-(2C4)-K-KDO) [38], we included this LPSA component in both our membrane systems. Calcium ions were added for neutralizing the LPSA, and these assembled peptide–membrane complexes were further subjected to the CHARMM36m force field [39]. Further, molecular dynamics simulations were performed using GROMACS 2021.5 [35]. Initial steric clashes in the systems were removed through energy minimization using the steepest descent method for 5000 steps, followed by 1 ns of NVT (constant number of particles, volume and temperature) and NPT (constant number of particles, pressure and temperature) equilibration. The velocity-rescaling thermostat [40] and Berendsen barostat [41] were employed for temperature and pressure control during the NVT and NPT equilibration phases, respectively. Lastly, the production MD simulations were conducted for a duration of 300 ns, and the subsequent analysis of the resulting trajectories was performed using Gromacs utilities. The MD simulation trajectory was visualized using Visual Molecular Dynamics (VMD) software (1.9.4) [42].

#### 2.7.3. Binding Energy Calculations

To estimate the binding affinity of G1 and G2 peptides with the lipid membrane, we employed the MM-GBSA method as implemented in the gmx_MMPBSA tool [43]. The binding energy calculations were performed on the last 100 ns (from 200–300 ns), and 1001 frames were extracted from the stable trajectory. The total binding energy for both systems will eventually help us understand the effectiveness of the peptides on the *C. muridarum* membrane. The entropy contribution was omitted during calculation due to computational cost, as in an earlier study [44].

### 2.8. Statistical Analysis

The data presented in this study involved three independent experiments (n = 3), which were repeated three *independent* times. Statistical analysis was performed using GraphPad Prism 10 software. A one-way analysis of variance (ANOVA) was used to determine the significance between the control and the experimental groups. In all figures, columns represent the mean of the data collected, and error bars represent SEM. Asterisks signify *p*-values: * < 0.05, ** < 0.005, *** < 0.0005 and **** < 0.0001.

## 3. Results

### 3.1. Expression of 3-O Sulfotransferase-3 (3-OST-3) and C. muridarum Entry

While the variable chains of heparan sulfate (HS)-containing glycosaminoglycans (GAGs) on the surfaces of mammalian cells play an important role in microbial invasion, the direct role of 3-O sulfated heparan sulfate (3-OS HS) as an independent determinant in *Chlamydia* cell infectivity remains unknown. Therefore, we tested the hypothesis that the presence of 3-OS HS is essential for chlamydial infection using a widely accepted and experimentally simple assessment in the CHO-K1 cell model. We used the wild-type CHO-K1 cells because they lack endogenous 3-OST-3 enzymes, which modify heparan sulfate (HS) to generate 3-OS HS receptors [33]. The expression of 3-OST-3 in CHO-K1 cells was confirmed by running the HSV-1 entry assay in parallel, since HSV-1 independently uses 3-OS HS to enter the CHO-K1 cell). Similarly, the role of the 3-OST-3 enzyme has been well-documented in mediating the entry of medically relevant viruses, including HSV-1 and CMV, into the host cell and virus cell-to-cell spread [23,45]. In this experiment, the CHO-K1 cells were transfected with the 3-OST-3 mammalian expression plasmid, while in parallel, CHO-K1 cells were transfected with an equal amount of an empty vector pCDNA3.1 (Figure 2A,B). Untransfected or mock transfected CHO-K1 cells were used as an internal control. The group of CHO-K1 cells expressing pCDNA3.1, 3-OST-3 and/or mock transfected control were then independently infected with *C. muridarum* at two different MOIs (0.1 and 0.25) for cell entry. As indicated in the Methods section, fluorescent-based staining followed by imaging to count the number of inclusion bodies was used as a parameter to define the cell infectivity of *C. muridarum*. There was no statistical difference in *C. muridarum* infectivity at the tested MOIs in the cells expressing the empty vector pCDNA3.1 control and or 3-OST-3 enzyme (Figure 2C,D). Taken together, our results indicate that the cell surface expression of 3-OS HS receptor does not serve an essential, integral function in promoting *C. muridarum* infection.

### 3.2. Preincubation of G1 and G2 Peptide Effects on Elementary Body (EB) Infectivity

Interestingly, previous findings suggest that heparin-like sulfated moieties are present on *Chlamydia* outer membrane protein (OMP) [46,47,48], while the cysteine-rich outer membrane polypeptide, OmcB, is a surface-exposed protein that functions as a chlamydial adhesin [49]. Published studies in the entry field suggest that the level and position of sulfation within HSPG play a role in binding, entry and inclusion formation [23,50,51]. Therefore, we next aimed to target a chlamydial infectious EB particle by directly exposing it to phage-display-derived G1 and G2 peptides recognizing heparan sulfate and 3-O sulfated HS ligand, respectively, before infecting the target HeLa (Figure 3) and VK2 cells (Figure 4). The details regarding the origin of the G1/G2 peptides are shown in Appendix A.

As previously mentioned, the formation of a membrane-bound compartment in the form of inclusion bodies was used as an indicator for Chlamydia’s cell infectivity in the presence and absence of the peptides. The pre-treatment of EBs with the G2 peptide ligand in a dosage-dependent manner showed a significant (*p* < 0.001) inhibition of *C. muridarum* infectivity of HeLa cells, as evident from the significant reduction in the green punctate, which corresponded to the number of inclusion bodies (Figure 3). The extent of inhibition over the range of G2 concentrations (0.00036–0.0029-μM) was similar, ranging from 80.2% to 88.5% inhibition, as compared to control, respectively. In contrast, preincubation of EBs with G1 peptide resulted in an enhancement in the number of inclusion bodies. Specifically, the EBs treated with 0.0016 μM of G1 peptide resulted in a 32% increase in infectivity into HeLa cells, while those with cell infection at the remaining concentrations (0.0007, 0.00036 and 0.0029 μM) were similar to that measured for the untreated EB control (Appendix A).

The impact of the G1 and G2 peptide ligand on the EBs’ pretreatment with infected vaginal epithelial cells (VK2 cells) was further determined to ascertain whether the EB neutralization effects were specific for HeLa cells. In addition, *C. muridarum,* being a murine pathogen similar to the human pathogen *C. trachomatis,* infects the vaginal epithelium [52]. Therefore, VK2 cells also served as an appropriate model to understand the effect of G1/G2 peptides targeting the HS and 3-OS HS receptors, improving the molecular basis of host–pathogen interactions. Interestingly, in contrast to our findings with HeLa cell infectivity, both the G1 and the G2 peptide ligands were highly effective in inhibiting *C. muridarum* infection in the VK2 cell model by significantly reducing the number of inclusion bodies in a dosage-dependent manner (Figure 4). Our results indicated that the G1 peptide ligand at the highest concentrations tested (0.003 and 0.0016 μM) significantly (*p* < 0.05) inhibited the EB infection of VK2 cells to 47% and 52% of the control, respectively. All of the G2 peptide ligand concentrations tested significantly (*p* < 0.001) inhibited *Chlamydia* cell infectivity by reducing the number of inclusion bodies  to that measured for untreated EBs control. This result indicates the significance of HS during EB’s infectivity in the two clinically relevant cell lines used in this study. Since the G1 and G2 peptide recognizes HS and 3-OS HS, it was logical to test whether the pre-treatment with the G1/G2 peptide for the host cells (HeLa and VK2) would have an impact on the infectivity of the *C. muridarum.* As indicated in Appendix A, it was clear that pre-incubation of the G1 and G2 peptides at the majority of the tested concentrations did not affect the entry of the EBs into the HeLa cells, but the G2 peptide did show some blocking effects at a higher concentration. Similarly, the peptide pre-treatment in the VK2 cells had some blocking effects at a higher concentration. Because 3-OS HS expression alone was not utilized as an independent receptor for Chlamydia entry (Figure 2), it was expected that blocking of 3-OS HS by G2 peptide would not be sufficient to abolish the entry of EBs into the host cell. This data also opens the possibility that *C. muridarum* utilizes several other receptors, which could vary depending on cell types.

### 3.3. Investigating the Interaction of G1 and G2 Peptides with the Chlamydia Lipid Bilayer

Iterative Threading ASSEmbly Refinement (I-TASSER) generated five potential 3D structures for each G1 and G2 peptide, ranking them by confidence C-score, a measure of model confidence. The best-fit models were selected based on C-score, template modeling (TM) score and root mean square deviation (RMSD) (Table 1). Model 1 was chosen as the most suitable structure for both peptides based on its C-score (the higher the better). G2 peptide exhibited a higher C-score than G1, suggesting improved folding and structural stability.

Molecular dynamics (MD) simulations were conducted to explore the interactions of the G1 and G2 peptides with the Chlamydia lipid bilayer, offering more profound insights into their binding modes and affinities (Figure 5B–E and Appendix A). Molecular dynamics simulations were performed using Gromacs 2021.5 [26]. The stability of the simulated complexes was assessed by analyzing the RMSD plot, as shown in Figure 6A. The RMSD plot indicates that the G2 peptide reaches equilibrium after 150 ns in the presence of the lipid bilayer, suggesting that it achieves both conformational and structural stability, unlike the G1 peptide (Appendix A). The G1 peptide exhibits significant conformational changes (Appendix A) and also achieves equilibrium after 150 ns, as can be observed on the RMSD plot (Figure 6A). To further explore the conformational properties of the G1 and G2 peptides, we calculated the root mean square fluctuations (RMSFs), shown in Figure 6B. The RMSF plot measures the movement degree of C_α_ atoms around their average positions. Here, the G2 peptide generally exhibits lower RMSF values across the majority of the residues compared to the G1 peptide (Figure 6B), and this suggests that the residues in the G2 peptide experience less movement and are more constrained. The regions around residue numbers 4–5 and 6–7 for the G2 peptide show particularly low RMSF values (below 2 Å in most cases), indicating that these parts of the peptide are highly stable and likely deeply embedded or strongly interacting with the lipid bilayer of the membrane (Figure 6B and Appendix A). Even where the RMSF values for G2 increase slightly (around residues 5 and 10–12), they generally remain lower than the corresponding residues in G1, suggesting a more consistently stable interaction profile across the entire peptide (Figure 6B). Conversely, the higher RMSF values observed for the G1 peptide indicate greater flexibility and weaker interactions with the membrane, with certain regions showing particularly high mobility (Figure 6B and Appendix A). Furthermore, the conformation and folding state of the G1 and G2 peptides were analyzed using the radius of gyration (Rg). The G2 peptide generally fluctuates at a lower average Rg value (around 7–8 Å) throughout the simulation time compared to the G1 peptide. This suggests that, on average, the G2 peptide maintains a more compact conformation during the simulation (Figure 6C). By contrast, the G1 peptide exhibits a higher average Rg value (around 8–9 Å and sometimes exceeding 9 Å) compared to the G2 peptide throughout the simulation. This indicates that the G1 peptide, on average, adopts a less compact and more extended conformation, as shown in Figure 6C. Even though G1 has weaker interactions with the membrane overall, it undergoes a transient conformational change after 150 ns that leads to a more compact structure (Figure 6C). This compaction might not necessarily indicate a stronger or more stable interaction with the membrane, but rather G1 might have become more compact while still primarily residing on the surface of the membrane or interacting only with the LPSA, rather than inserting deeply into the hydrophobic core, like G2, as can be observed in Appendix A.

To examine how these conformational effects influence binding affinity, we conducted binding energy calculations using the gmx_MMPBSA tool [32].

### 3.4. Binding Energy Calculations

To assess the binding affinity of G1 and G2 peptides with the lipid bilayer, we performed binding energy calculations using the MM-GBSA method through gmx_MMPBSA tool [32]. The binding energy data revealed that the G2 peptide had a higher affinity (−111.76 kcal/mol) with the lipid bilayer compared to the G1 peptide with the lipid bilayer (−40.61 kcal/mol), as shown in Table 2. Moreover, the G1- and G2-peptide-associated lipid bilayer complexes reveal negative van der Waals and electrostatic energy, indicating favorable interactions. The G1 peptide shows slightly higher van der Waal energy compared to the G2 peptide, but the G2 peptide has much stronger electrostatic interactions, suggesting that it has more charged or polar regions interacting with the membrane. Furthermore, the G2 peptide has a higher desolvation penalty (ΔE_GB_ = 36001.98), reflecting its stronger initial solvation before interacting with the membrane as compared to the G1 peptide. Also, the G2 peptide with a more negative value (ΔG_GAS_ = −36104.32) likely has a more hydrophobic surface exposed to the membrane as compared to the G1 peptide. The ΔG_GAS_ values indicate favorable interactions in the gas phase, with the G2 peptide showing stronger interactions as compared to the G1 peptide. The interaction is more energetically favorable than the G1 peptide, with a total free energy of −111.76 kcal/mol. This is largely due to stronger electrostatic interactions, indicating a more stable or tighter binding with the membrane. The higher solvation energy value reflects the initial state of the peptide being well-solvated before binding. Taken together, the G2 peptide–membrane interactions are more stable and energetically favorable than in the G1 membrane complex (Table 2 and Appendix A), primarily due to stronger non-bonded interactions. This suggests that the G2 peptide has more extensive or effective interactions with the membrane, which may be important for its function or stability in a biological context.

## 4. Discussion

Infection of host cells by *Chlamydia* is a multi-step dynamic process [50,51,52,53,54]. It is widely believed that *C. muridarum* first mediates a low-affinity interaction with HSPGs [6,8], followed by interacting specifically with high-affinity binding to host cell receptors. These receptors include a wide variety of proteins, ranging from cystic fibrosis transmembrane conductance regulator (CFTR), mannose 6-phosphate receptor (M6P), β1 integrin, epidermal growth factor receptor (EGFR), receptor tyrosine kinases, fibroblast growth factor (FGF) receptor and its ligand, FGF, platelet-derived growth factor receptor (PDGFR) and protein disulfide isomerase (PDI), a component of the estrogen receptor complex [9,55,56,57,58,59,60,61,62].

In this study, we assessed whether 3-O sulfation in the HS chain is preferentially utilized by *C. muridarum* during cell entry. Identifying the key sulfated residues involved during *Chlamydia* infection is an important step to better understand cell- and tissue-specific requirements towards the development of novel treatments. We first deployed a well-established CHO-K1 cell-based assay, which was adapted from the HSV-1 entry field to test the significance of 3-O sulfated heparan sulfate in *Chlamydia* infectivity. Our entry data with CHO-K1 cells expressing either empty pCDNA3.1 vector or 3-OST-3 enzyme showed no differences in the infectivity of *C. muridarum*, suggesting that 3-OS HS does not serve as an independent entry receptor. This result contrasts with previously established HSV entry data, according to which viruses use 3-OS HS to promote viral entry and cell-to-cell spread [23,33]. This data provided an early presumptive, yet critical clue that two independent pathogens such as HSV and Chlamydia perhaps do not compete for the same 3-OS HS receptor during either co- or super-infection. The expression of HS and 3-OS HS receptors is well-documented in multiple susceptible cell lines used by STI (sexually transmitted infection) pathogens, including dendritic cells, T helper cells and the genital epithelium [15,63,64]. Therefore, a preferential recognition of specific 3-OS HS receptors by HSV-1 glycoprotein D (gD) functions as a key determinant to allow the potential occurrence of HSV infection [16], while Chlamydia’s ability to use multiple cell surface receptors may provide unique flexibility to enter cells along with HSV. However, future studies will be needed to understand the significance of sulfated HS in a co- or super-infection model. Although the 3-OST-3-generated 3-OS HS receptor did not enhance *C. muridarum* cell infectivity, in the future, it will be worth investigating whether expression of other 3-OST enzymatic isoforms, such as 3-OST-1, -2, -4, -5 and -6, would have any impact on the infectivity of *C. muridarum* and, possibly, any associated inflammatory damage [45]. Similarly, other host genes, such as exostosin-1 (EXT-1) and N-deacetylase/N-sulfotransferase (NDST), which serve as critical players in HS biosynthesis, would also be interesting to test for their modulatory role in Chlamydia cell infectivity. Interestingly, a recent finding highlighted that EXT-1 plays a unique role in the life cycle of Zika virus infection [65]. Knocking down EXT1 at the early stage represses viral infection, whereas it increases virus infection when knockdown is performed at the late stage of viral infection [65]. In addition, since the HS chain is also modified by multiple isoforms of 2-O sulfotransferases (2-OST-1 and 2-OST-2) and 6-O sulfotransferases (6-OST-1, 6-OST-2 and 6-OST-3) before 3-O sulfotransferases (3-OST) [13], it remains unknown whether 2-OSTs and or 6-OSTs independently participate in the infectivity of *C. muridarum.* A previous study reported that the specific removal of 6-O sulfation by human endosulfatases diminished *C. muridarum* binding and decreased vacuole formation in the HeLa cell model, suggesting that 6-O sulfation is indeed a critical determinant of *C. muridarum* [22].

Our assessment of the impact of peptides targeting HS (G1) and 3-O sulfated HS (G2) on *C. muridarum* infectivity generated interesting findings. For instance, the pre-exposure of the G2 peptide to *Chlamydia* resulted in a significant blockage of infection in both the relevant cell lines used in this study. The above results are complemented by a previous finding that showed a heparan sulfate-specific monoclonal antibody that binds to glycosaminoglycans (GAGs) localized on the surface of the chlamydial organism and effectively neutralizes the infectivity of both *C. trachomatis* and *C. pneumoniae* [46]. Evidence of sugar being present on the outer surfaces of multiple other intracellular pathogens is also documented. For instance, *Toxoplasma gondii* displayed lectins and glycoconjugates on their surfaces [66]. In addition, the rhoptry and dense granule protein of Toxoplasma binding recognizes both N-sulfation and 6-O-sulfation on the glucosamine residue, with N-sulfation seeming to be more important than *O*-sulfation [67]. Similarly, the G protein of respiratory syncytial virus (RSV) possesses a short chain of oligosaccharide structures, which are involved in its infectivity in vitro [68]. Whether Toxoplasma and RSV carry the 3-O sulfation-like moieties on their surfaces remains to be determined. One major benefit of the presence of HS on various pathogens, including *Chlamydia,* is that it allows the use of ubiquitous multiple receptors for attachment and invasion. Furthermore, the use of HS is also advantageous because it presents a host-similar ligand on its surface, allowing it to exploit molecular mimicry and thereby prevent recognition by the host immune system [46]. However, the masking of EB ligands by HS may alter the display of alternative attachment ligands, which could alter infectivity in a manner that is host-cell-specific.

Interestingly, pre-treatment of *Chlamydia* with the G1 peptide uniquely promoted cell infectivity in HeLa cells, which was unusual. Since G1 peptide possesses affinity for HS, it is quite possible that sequestration or binding of HS by the G1 peptide resulted in an open state conformation of the sulfated moieties present on the HS chain, which provided easy access to the sulfated domains to promote the entry of *C. muridarum*. However, we did not notice a similar promoting effect of the G1 peptide in the VK2 cells, which could have been due to the differential expression of HS and the enzymatic isoforms of 3-OST-3. Another possibility is that G1 and G2 peptides also appear to modulate HS expression [69,70]. Similarly, other pathogens, such as HSV-2, have been shown to enhance the expression of HS and 3-OS HS upon exposure to the host cell, resulting in a higher affinity for the G2 peptide in virally infected cells compared to uninfected cells [32]. Further studies are needed to understand the glycans-based differences among cell types, as well as the independent effects of the peptides or *Chlamydia* exposure on HS/3-OS HS expression, which may influence *Chlamydia*–host cell interactions.

Taken together, our data demonstrate that the blocking of 3-OS HS moieties by the G2 peptide mediates the neutralization of *C. muridarum* cell infectivity. Based on our findings with the G2–lipid bilayer binding affinity analysis, we propose that the peptides targeting the lipid bilayer may have affected the cell dynamics and were able to prevent *Chlamydia* cell infection (Figure 7A,B) by having strong affinity towards the lipid membrane (Figure 7C). It is quite possible that the latter may also pose the 3-O sulfated HS-like domains on the polysaccharide, which could be the cause of the neuralization effect observed during the EBs’ exposure to the G2 peptide. Since the human strains of *Chlamydia (C. trachomatis)* and the associated serovars are known utilizers of a diverse subset of sulfated HS to promote cell entry [71], future studies will be needed to address whether the G2 peptide-mediated EB neutralization effect persists in other serovars, including assessing the impact of G1/G2 peptides on Chlamydia–HSV-associated co-infection and or super-infection in the murine model.

## 5. Conclusions

Our findings demonstrate that G2 peptide, which recognizes 3-O sulfation in HS, significantly inhibits *C. muridarum* entry into host cells, suggesting a strong interaction between the peptide and components of the chlamydial EBs. Molecular dynamics simulations further suggest that the G2 peptide interacts with the lipid bilayer, thereby disrupting the structural integrity and infective dynamics of *C. muridarum*. Collectively, these results emphasize the critical role of sulfated glycoconjugates in chlamydial pathogenesis and highlight them as promising targets for therapeutic intervention. Future studies should explore the serovar-specific nature of this interaction and evaluate its implications in co-infections with herpes simplex virus (HSV) and other sexually transmitted pathogens.

## Figures and Tables

**Figure 1 biomolecules-15-00999-f001:**
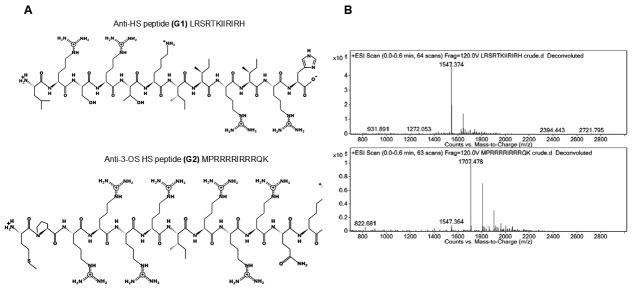
Characterization of the 12-mer HS (G1) peptide ligand and 3-OS HS (G2) peptide ligand. The sequence of HS (G1) peptide ligand and 3-OS HS (G2) peptide ligand (panel **A**), including the mass spec analysis of the purified peptides is shown (panel **B**). Purification of G1 and G2 peptides was achieved using preparative scale reverse-phase high-performance liquid chromatography (Waters Prep 150) with a Phenomenex Kinetex column (C-18 stationary phase, 5 μm, 100 Å pore size, 30 × 150 mm). The mobile phase consisted of acetonitrile and water with 0.1% trifluoroacetic acid (TFA). Fractions containing the desired peptides were confirmed using electrospray ionization mass spectrometry (ESI-MS) on an Agilent 6520 Quadrupole Time-of-Flight (Q-ToF) via direct injection. MassHunter Workstation Data Acquisition software operated the instrument, and MassHunter Qualitative Analysis software handled data analysis and processing. Excess acetonitrile was removed from pooled fractions by rotary evaporation. Samples were then freeze-dried and the resulting powders were stored at −20 °C. Purity was confirmed using liquid chromatography–mass spectrometry (LC-MS) on an Agilent 1200 system with a Phenomenex Jupiter C-12 column (1 × 100 mm; 4 µm). The mass detector (MS) was an Agilent 6520 Q-TOF MS.

**Figure 2 biomolecules-15-00999-f002:**
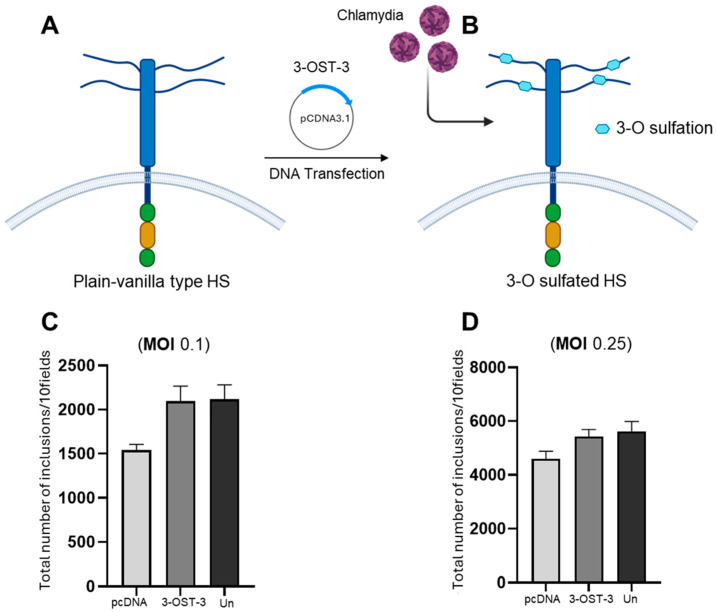
The impact of 3-O sulfotransferase-3 (3-OST-3) isoform on *C. muridarum infectivity*. (Panel **A**) shows that wild-type CHO-K1 cells, which lack endogenous HS modifying enzymes, were transfected with the human encoded 3-OST-3 isoform before infection with *C. muridarum*. The image in (Panels **A**,**B**) was created by BioRender. (Panel **C**,**D**) show the dosage-dependent (0.1 and 0.25 MOI) effect of *C. muridarum* on CHO-K1 cell infectivity in the presence and absence of the 3-OST-3 enzyme. Data is representative of the results of three independent experiments (n = 3) performed in triplicate.

**Figure 3 biomolecules-15-00999-f003:**
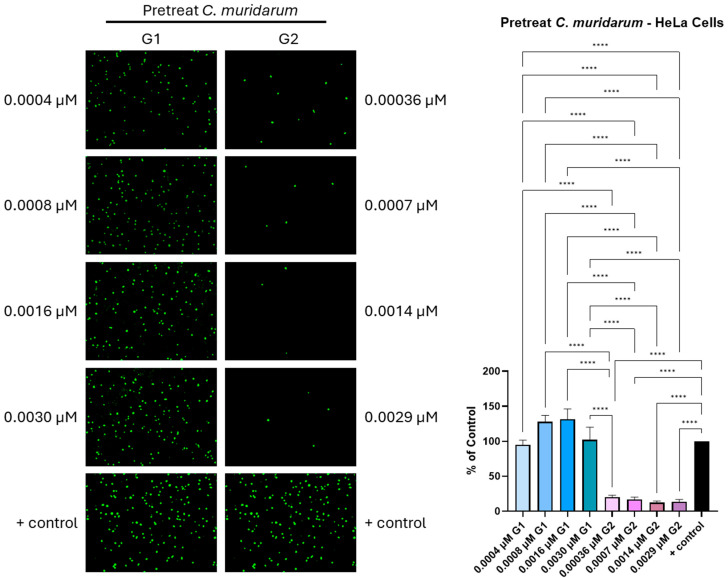
Effect of the pretreatment with various concentrations of G1 and G2 peptides on *C. muridarum* infection of HeLa cells, as determined by fluorescent microscopy. *C. muridarum* was pre-treated with G1 or G2 peptides before infecting HeLa cells. Inclusions were stained, and representative images are shown. The number of inclusions was counted and the percentage of infected cells treated with G1 or G2 compared to non-treated infected cells only (positive control) was calculated (Mean ± SEM, **** *p* < 0.0001). Data is representative of the results of three independent experiments (n = 3) performed in triplicates.

**Figure 4 biomolecules-15-00999-f004:**
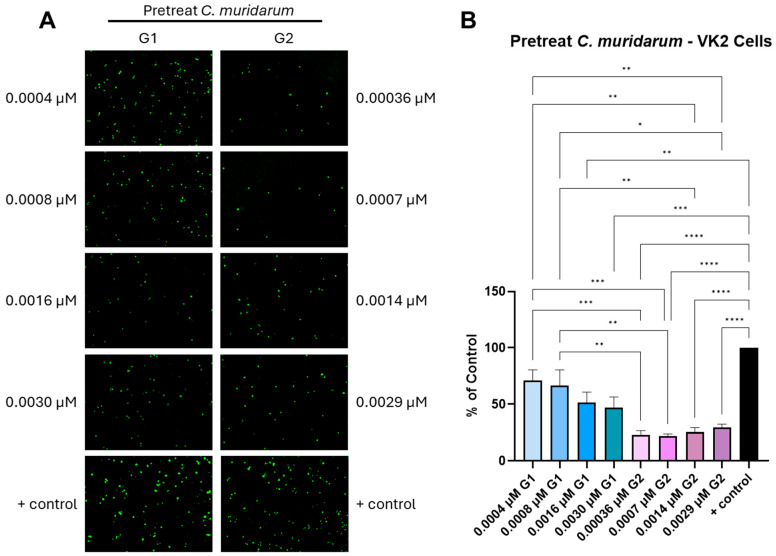
(**A**) Effect of the pretreatment with various concentrations of G1 and G2 peptides on *C. muridarum* infection of VK2 cells, as determined by fluorescent microscopy. *C. muridarum* was pre-treated with G1 or G2 peptides before infecting VK2 cells. Inclusions were stained and representative images were shown. (**B**) The number of inclusions was counted and the percentage of infected cells treated with G1 or G2 compared to non-treated infected cells only (positive control) was calculated (Mean ± SEM, * *p* ≤ 0.05, ** *p* < 0.005, *** *p* < 0.0005 and **** *p* < 0.0001). Data is representative of the results of three independent experiments (n = 3) performed in triplicates.

**Figure 5 biomolecules-15-00999-f005:**
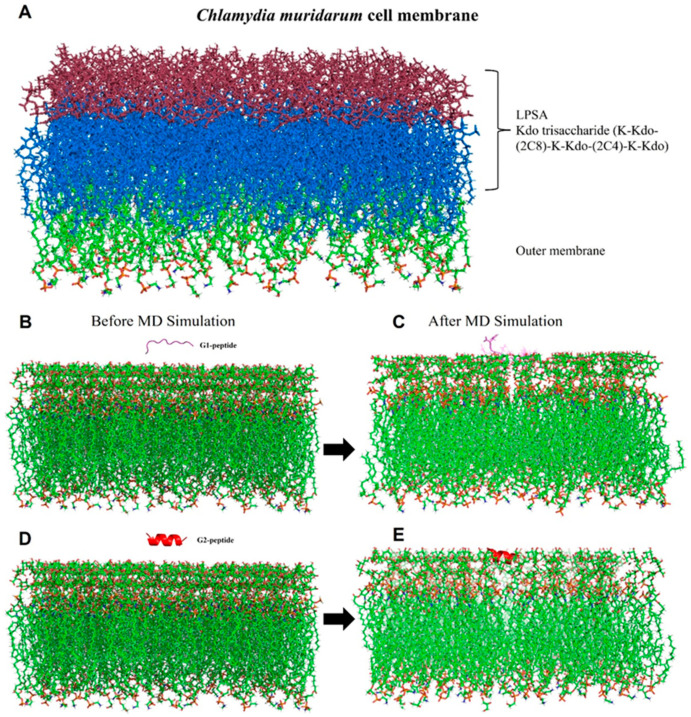
Molecular dynamics (MD) simulation of Chlamydia muridarum lipid bilayer and its interaction with G1 and G2 peptides. The interaction of G1 (purple) and G2 (red) peptides with the *C. muridarum* outer membrane is analyzed. (**A**) shows the representation of Chlamydia muridarum lipid bilayer building and simulation protocol, (**B**) shows the *C. muridarum* outer membrane composed of LPS-A. (**C**,**E**) Initial configurations of G1 and G2 peptides placed 20 Å above the membrane surface. (**D**,**E**) Post-MD simulation snapshots showing the peptide–membrane interactions, where G2 exhibits stable membrane association and deeper interaction compared to G1.

**Figure 6 biomolecules-15-00999-f006:**
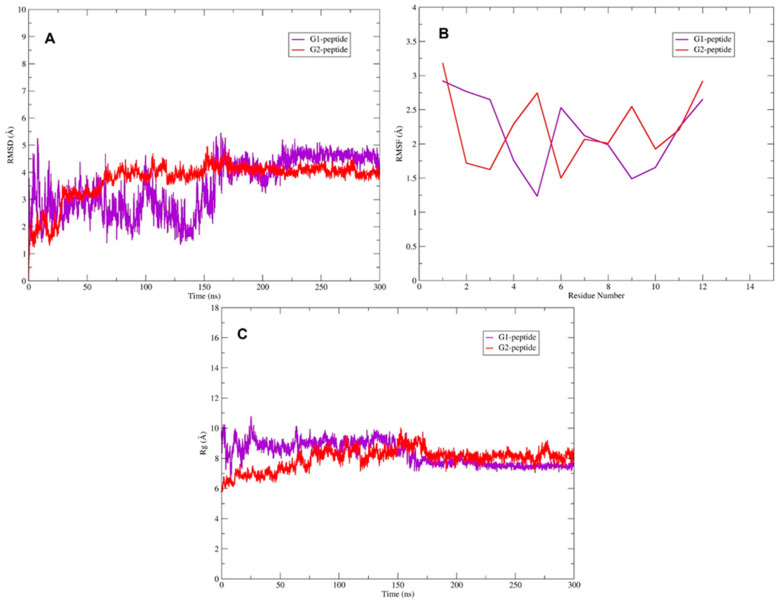
Molecular dynamics (MD) simulation analysis of *Chlamydia muridarum* in complex with G1 and G2 peptides. The quantitative interactions of G1 (purple) and G2 (red) peptides with the *C. muridarum* membrane are illustrated. (**A**) Root Mean Square Deviation (RMSD) plot indicates that the G2 peptide achieves structural stability around the 150 ns time point, showing greater stability than G1. (**B**) Root Mean Square Fluctuation (RMSF) plot reflects residue-level flexibility. (**C**) Radius of gyration (Rg) analysis reveals that G2 maintains a more compact, folded conformation within the lipid membrane over 300 ns MD simulation.

**Figure 7 biomolecules-15-00999-f007:**
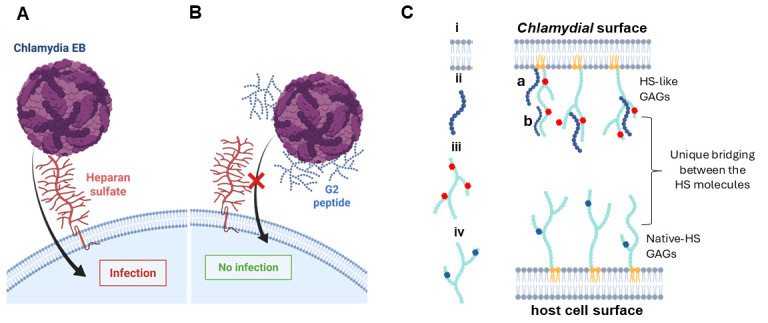
Significance of 3-O sulfation in *C. muridarum* entry. (**A**) *C. muridarum* infections occur in the absence of G2 peptide, (**B**) while a significant blockage in cell entry was noticed when *C. muridarum* was exposed to G2 peptide (which targets 3-O sulfation) before cell entry. (**C**). Proposed model for G2-mediated inhibition during cell entry of *Chlamydia muridarum*. Based on computational structural modeling data, the G2 peptide poses a significantly higher affinity towards the *C. muridarum* lipid membrane (a). It is also possible that the G2 peptides may also recognize the glycosaminoglycan (GAG)-like molecules with 3-O sulfation activity present on the OMP (b). Having GAG-like sulfated moieties on polysaccharides on the chlamydial surface along with HS-GAGs on the host cell provides critical bridging interactions between similar molecules, which offer a distinct advantage to *Chlamydia* in invading the immune system and remain persistent in the host. (i) OMP, (ii) G2 peptide, (iii) presence of 3-O sulfated moieties on OMP, (iv) native 3-O sulfation on the host cell. This image was created by BioRender. Tiwari V (2025) https://app.biorender.com/illustrations/canvas-beta/67f5cee0498da73002668e5f (accessed on 21 April 2025).

**Table 1 biomolecules-15-00999-t001:** Selection parameters of the I-TASSER server with the best 3D structure of G1 and G2 peptides.

Parameters	G1 Peptide	G2 Peptide
Confidence score (C-score)	−1.02	0.12
Template modelling score (TM score)	0.59 ± 0.14	0.73 ± 0.11
Root mean square deviation (RMSD)	1.9 ± 1.6 Å	0.5 ± 0.5 Å

**Table 2 biomolecules-15-00999-t002:** Binding energy contribution of G1 and G2 peptides with *C. muridarum* lipid bilayer. All energies are given in kcal/mol. VDW = Van der Waals, ELE = Electron binding energy, GB = Generalized Born, GAS = Change in Gibbs free energy (ΔG), SURF = Surface binding energy, SOLV = Solvation.

Peptide–Membrane Complexes	ΔE_VDW_	ΔE_ELE_	ΔE_GB_	ΔE_SURF_	ΔG_GAS_	ΔG_SOLV_	ΔTotal
G1 membrane	−44.66	−20,879.97	20,893.80	−9.78	−20,924.63	20,884.02	**−40.61**
G2 membrane	−26.62	−36,077.70	36,001.98	−9.42	−36,104.32	35,992.55	−111.76

## Data Availability

The original contributions presented in this study are included in the article. Further inquiries can be directed to the corresponding authors.

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
