# Peer review of "3-O Sulfated Heparan Sulfate (G2) Peptide Ligand Impairs the Infectivity of Chlamydia muridarum"

_biomolecules, 2025, doi:10.3390/biom15070999_

Round 1
Reviewer 1 Report
Comments and Suggestions for Authors
The manuscript presents an in vitro and in silico study of the effect of HS 3-O sulfation on the infectivity of Chlamidia muridarum. I find that the paper has many issues that require at least a major revision.
1. I do not agree whatsoever with the authors' conclusions in the Abstract (lines 28–30) that their ”results highlight the significance of 3-O sulfated heparan sulfate and other GAGs involved during the entry of C. muridarum into host cells.”
2. Lines 57–60: These are proteoglycans, and not GAGs. GAGs can be attached to core proteins to form PGs, but can also exist as free polymers, e.g. hyaluronic acid or heparin.
3. Some mention of the G1 and G2 peptides in the Introduction is necessary. What are these peptides? What evidence is there that they selectively bind HS and 3-OS HS specifically? Both peptides are highly positively charged, so I bet they would bind with high affinity to any heparin/HS chain, regardless of specific sulfation patterns.
4. Section 2.5.2: What was the specific composition of the Gram-negative membrane used? The authors need to provide detailed information on their computational model.
5. Section 2.5.2: Was only the outer membrane used, or the whole bacterial envelope?
6. Section 2.5.2: I assume that since the CHARMM-GUI server was used for input generation, the force field for the MD simulations was CHARMM 36? Please, list your protocol in detail!
7. Lines 270–273 belong to the Materials and Methods section.
8. Lines 279–182: In general, 100 ns are highly insufficient to sample properly conformational changes in the peptides. In addition, when dealing with membranes, one expects hundreds or even thousands of ns of simulation time. I understand that simulation lengths depend on available computational resources. But to claim that 30 ns are sufficient for reaching both structural and conformational equilibrium is just ridiculous.
9. In general, the observed RMSD variations are not unexpected and are not suggestive of some large differences between the two peptides.
10. The Rgs of the two peptides differ, because the authors' input models are different – one is fully folded into a helix, and the other one is rather disordered. Again, the chosen simulation time frame does not allow for deep conclusions based on this metric.
11. Lines 304–308: Comparing distributions between two independent PCAs of two different systems does not make any sense. You cannot extract any useful information from Fig. 6D. I suggest the authors try to understand what PCA is as a mathematical concept.
12. Section 3.4: So, these are MM-GBSA calculations from a single trajectory, is that right? You already have a rather limited sampling of the peptide—membrane interaction of just 100 ns. Relying on MM-GBSA calculations from a single trajectory can introduce significant artifacts, as it neglects conformational differences between the bound and unbound states; using separate trajectories for the complex, free peptide, and free membrane would better capture the distinct energetic contributions and would yield a more thermodynamically rigorous estimate of binding free energy.
13. Fig. 7C: If GAGs on the bacterial envelope bind to GAGs on the host cells, wouldn’t it make sense to conduct in vitro experiments of when host cells are pretreated with G1, G2, or HS to check if they would competitively inhibit attachment of the EBs?
14. Please, include a list of abbreviations for better readability.
In summary, the paper is not yet at a stage suitable for publication and would benefit from significant improvement.
Author Response
Reviewer: 1
Comments to the Author
The manuscript presents an in vitro and in silico study of the effect of HS 3-O sulfation on the infectivity of Chlamydia muridarum. I find that the paper has many issues that require at least a major revision. In summary, the paper is not yet at a stage suitable for publication and would benefit from significant improvement.
This reviewer raised multiple good points which strengthened our findings. We agree with the reviewer’s comment and hence we have now provided substantial evidence by clarifying all the details throughout the manuscript.
Major revisions:
- I do not agree whatsoever with the authors' conclusions in the Abstract (lines 28–30) that their” results highlight the significance of 3-O sulfated heparan sulfate and other GAGs involved during the entry of muridarum into host cells.
Response: As per the reviewer’s recommendation, we have now carefully restated the conclusion in the abstract section that 3-O sulfated heparan sulfate recognizing G2 peptide prevents the entry of C. muridarum.
- Lines 57–60: These are proteoglycans and not GAGs. GAGs can be attached to core proteins to form PGs, but can also exist as free polymers, e.g. hyaluronic acid or heparin.
Response: As per the reviewer’s comment we have now corrected the statement on page of the revised manuscript.
- Some mention of the G1 and G2 peptides in the Introduction is necessary. What are these peptides? What evidence is there that they selectively bind HS and 3-OS HS specifically? Both peptides are highly positively charged, so I bet they would bind with high affinity to any heparin/HS chain, regardless of specific sulfation patterns.
Response: We have now discussed the origin and significance of G1/G2 peptides in the Introduction section in detail on page. We have also explained and cited our previous studies which have shown that removal of sulfated HS by heparinase III treatment results in significant loss of recognition of G2 peptide to cell surface sulfated HS (Ali et al., 2012 Journal of Virology).
We agree with the reviewer’s comments that due to the cationic nature of G1 and G2 peptides they may possibly interact with other heparin chains as well. Hence, we have now clearly stated that beside recognition of G2 peptide to HS/3-OS HS sites, the affinity of G2 peptide to other sites in HS chains or other pathogenic proteins can’t be ignored which may further be contributing towards the hindering the entry of Chlamydia muridarum. Since G1 and G2 peptides also appear to modulate HS expression (Park et al., 2013 IOVS) future studies will be helpful if the peptide exposure may influence Chlamydia–host cell interaction by gain or loss of HS expression. We have now discussed all the above possibilities in detail in the discussion section in the revised manuscript.
- Section 2.5.2: What was the specific composition of the Gram-negative membrane used? The authors need to provide detailed information on their computational model.
Response: Thank you very much for the suggestion. The lipid bilayer composition of the C. muridarum membrane, a Gram-negative bacterial pathogen, and the outer membrane of Chlamydia species contains a truncated LPS (long form) antigen with a group-specific epitope composed of a KDO trisaccharide (K-KDO-(2C8)-K-KDO-(2C4)-K-KDO). The inner leaflet of our model incorporated a mixture of PE, PG, and CL lipid headgroups in a 75:20:5 ratio. The information was obtained from the OPM database. We included these components in both our membrane systems. Moreover, these assembled peptide-membrane complexes were further used for MD simulations using Gromacs 2021.5 software. This information has been incorporated in the revised manuscript.
- Section 2.5.2: Was only the outer membrane used, or the whole bacterial envelope?
Response: In this study, the focus was primarily on the outer membrane, as Chlamydial outer membrane proteins (OMPs) play a crucial role in host cell interactions, primarily facilitating adhesion and entry.
- Section 2.5.2: I assume that since the CHARMM-GUI server was used for input generation, the force field for the MD simulations was CHARMM 36? Please, list your protocol in detail!
Response: We thank the reviewer for their thorough reading and the valuable suggestion to include a detailed protocol. Regarding the force field, we chose CHARMM36m due to its enhanced accuracy for simulating biomolecules, especially proteins and lipids, as it's a well-regarded refinement of the CHARMM36 all-atom additive force field. This information has been added to the revised manuscript for clarity.
- Lines 270–273 belong to the Materials and Methods section.
We have taken out the lines and moved it to material and method section.
- Lines 279–182: In general, 100 ns are highly insufficient to sample properly conformational changes in the peptides. In addition, when dealing with membranes, one expects hundreds or even thousands of ns of simulation time. I understand that simulation lengths depend on available computational resources. But to claim that 30 ns are sufficient for reaching both structural and conformational equilibrium is just ridiculous.
Response: We thank the reviewer for their critical assessment of the simulation length. We agree that 100 ns is a relatively short timescale for exploring the conformational landscape of peptides and their interactions with membranes, and we understand the preference for much longer simulations. As the reviewer correctly points out, simulation length is often constrained by available computational resources. However, we have taken the reviewer's feedback seriously and have extended the simulation time to 300 ns to ensure adequate sampling of the peptide conformational changes and membrane interactions. Based on the analysis of root-mean-square deviation (RMSD) as well as other relevant parameters, we observed that the system reaches structural and conformational equilibrium after approximately 150 ns. We have carefully re-evaluated our analyses and updated the manuscript with the findings from this extended simulation.
- In general, the observed RMSD variations are not unexpected and are not suggestive of some large differences between the two peptides.
Response: We agree with the reviewer that the observed RMSD variations are not indicative of major structural differences between the two peptides. However, the peptides differ in their amino acid composition and net charge, which may influence their interaction dynamics with the membrane. Our intention is to explore how these differences affect peptide stability and membrane binding, and to further assess their binding affinities. This computational analysis is aimed at complementing our experimental findings and providing additional insights into their mechanism of action. The updated RMSD plot revealed significant differences between the two systems, and the information is incorporated in revised manuscript.
- The Rgs of the two peptides differ, because the authors' input models are different – one is fully folded into a helix, and the other one is rather disordered. Again, the chosen simulation time frame does not allow for deep conclusions based on this metric.
Response: We appreciate the reviewer’s observation. Indeed, the initial structural differences in the input models-one being predominantly helical and the other more disordered-do contribute to the variation observed in the radius of gyration (Rg). We acknowledge that the original simulation time frame was not sufficient to capture the complete conformational convergence. However, with the extended 300 ns simulations and based on the MD simulation movies, the peptides folding has significantly differed, and we have revised the relevant section in the manuscript to reflect this clarification and have interpreted the Rg data with appropriate caution.
- Lines 304–308: Comparing distributions between two independent PCAs of two different systems does not make any sense. You cannot extract any useful information from Fig. 6D. I suggest the authors try to understand what PCA is as a mathematical concept.
Response: Thank you for your insightful comment. We acknowledge that comparing principal component distributions from two independent PCA analyses of different systems is not methodologically appropriate, as it does not yield meaningful or interpretable information. We appreciate the reviewer’s clarification and have removed this comparison and the associated figure (Fig. 6D) from the revised manuscript accordingly.
- Section 3.4: So, these are MM-GBSA calculations from a single trajectory, is that right? You already have a rather limited sampling of the peptide—membrane interaction of just 100 ns. Relying on MM-GBSA calculations from a single trajectory can introduce significant artifacts, as it neglects conformational differences between the bound and unbound states; using separate trajectories for the complex, free peptide, and free membrane would better capture the distinct energetic contributions and would yield a more thermodynamically rigorous estimate of binding free energy.
Response: We thank the reviewer for highlighting the limitations of our MM-GBSA calculations from a single trajectory. Yes, the original MM-GBSA binding energy calculations were performed using a single trajectory approach based on 100 ns simulation data. We agree with the reviewer that this approach has limitations, especially given the short simulation time and the neglect of conformational differences between the bound and unbound states.
In response to additional suggestions, we have extended the simulations to 300 ns, and for the MM-GBSA calculations, we have now used snapshots from the most equilibrated portion of the trajectory (i.e., the last 100 ns, from 200 to 300 ns). While we acknowledge that using separate trajectories for the complex, free peptide, and free membrane would provide a more rigorous thermodynamic estimate, the single-trajectory approach is widely used and accepted in membrane-binding studies due to its computational efficiency and consistency in sampling.
Due to limited computational resources, multiple independent simulations for the unbound components were not feasible in our case. However, we believe that the updated analysis using the extended simulation time significantly improves the reliability of the binding energy estimates. The revised results have been included in the updated manuscript. We will prioritize performing these separate simulations in future work to further validate our findings.
- 7C: If GAGs on the bacterial envelope bind to GAGs on the host cells, wouldn’t it make sense to conduct in vitro experiments of when host cells are pretreated with G1, G2, or HS to check if they would competitively inhibit attachment of the EBs?
Response: As per the reviewer’s suggestions, we have now provided the data on when G1 and G2 peptides were pre-treated to the host HeLa and VK2 cells before challenging with Chlamydia. This data has been added as a supplementary figure 2 in the revised manuscript.
- Please, include a list of abbreviations for better readability.
Response: As per the reviewer’s suggestions we have now included the list of abbreviations on page in the revised manuscript.
Reviewer: 1
Comments to the Author
The manuscript presents an in vitro and in silico study of the effect of HS 3-O sulfation on the infectivity of Chlamydia muridarum. I find that the paper has many issues that require at least a major revision. In summary, the paper is not yet at a stage suitable for publication and would benefit from significant improvement.
This reviewer raised multiple good points which strengthened our findings. We agree with the reviewer’s comment and hence we have now provided substantial evidence by clarifying all the details throughout the manuscript.
Major revisions:
- I do not agree whatsoever with the authors' conclusions in the Abstract (lines 28–30) that their” results highlight the significance of 3-O sulfated heparan sulfate and other GAGs involved during the entry of muridarum into host cells.
Response: As per the reviewer’s recommendation, we have now carefully restated the conclusion in the abstract section that 3-O sulfated heparan sulfate recognizing G2 peptide prevents the entry of C. muridarum.
- Lines 57–60: These are proteoglycans and not GAGs. GAGs can be attached to core proteins to form PGs, but can also exist as free polymers, e.g. hyaluronic acid or heparin.
Response: As per the reviewer’s comment we have now corrected the statement on page of the revised manuscript.
- Some mention of the G1 and G2 peptides in the Introduction is necessary. What are these peptides? What evidence is there that they selectively bind HS and 3-OS HS specifically? Both peptides are highly positively charged, so I bet they would bind with high affinity to any heparin/HS chain, regardless of specific sulfation patterns.
Response: We have now discussed the origin and significance of G1/G2 peptides in the Introduction section in detail on page. We have also explained and cited our previous studies which have shown that removal of sulfated HS by heparinase III treatment results in significant loss of recognition of G2 peptide to cell surface sulfated HS (Ali et al., 2012 Journal of Virology).
We agree with the reviewer’s comments that due to the cationic nature of G1 and G2 peptides they may possibly interact with other heparin chains as well. Hence, we have now clearly stated that beside recognition of G2 peptide to HS/3-OS HS sites, the affinity of G2 peptide to other sites in HS chains or other pathogenic proteins can’t be ignored which may further be contributing towards the hindering the entry of Chlamydia muridarum. Since G1 and G2 peptides also appear to modulate HS expression (Park et al., 2013 IOVS) future studies will be helpful if the peptide exposure may influence Chlamydia–host cell interaction by gain or loss of HS expression. We have now discussed all the above possibilities in detail in the discussion section in the revised manuscript.
- Section 2.5.2: What was the specific composition of the Gram-negative membrane used? The authors need to provide detailed information on their computational model.
Response: Thank you very much for the suggestion. The lipid bilayer composition of the C. muridarum membrane, a Gram-negative bacterial pathogen, and the outer membrane of Chlamydia species contains a truncated LPS (long form) antigen with a group-specific epitope composed of a KDO trisaccharide (K-KDO-(2C8)-K-KDO-(2C4)-K-KDO). The inner leaflet of our model incorporated a mixture of PE, PG, and CL lipid headgroups in a 75:20:5 ratio. The information was obtained from the OPM database. We included these components in both our membrane systems. Moreover, these assembled peptide-membrane complexes were further used for MD simulations using Gromacs 2021.5 software. This information has been incorporated in the revised manuscript.
- Section 2.5.2: Was only the outer membrane used, or the whole bacterial envelope?
Response: In this study, the focus was primarily on the outer membrane, as Chlamydial outer membrane proteins (OMPs) play a crucial role in host cell interactions, primarily facilitating adhesion and entry.
- Section 2.5.2: I assume that since the CHARMM-GUI server was used for input generation, the force field for the MD simulations was CHARMM 36? Please, list your protocol in detail!
Response: We thank the reviewer for their thorough reading and the valuable suggestion to include a detailed protocol. Regarding the force field, we chose CHARMM36m due to its enhanced accuracy for simulating biomolecules, especially proteins and lipids, as it's a well-regarded refinement of the CHARMM36 all-atom additive force field. This information has been added to the revised manuscript for clarity.
- Lines 270–273 belong to the Materials and Methods section.
We have taken out the lines and moved it to material and method section.
- Lines 279–182: In general, 100 ns are highly insufficient to sample properly conformational changes in the peptides. In addition, when dealing with membranes, one expects hundreds or even thousands of ns of simulation time. I understand that simulation lengths depend on available computational resources. But to claim that 30 ns are sufficient for reaching both structural and conformational equilibrium is just ridiculous.
Response: We thank the reviewer for their critical assessment of the simulation length. We agree that 100 ns is a relatively short timescale for exploring the conformational landscape of peptides and their interactions with membranes, and we understand the preference for much longer simulations. As the reviewer correctly points out, simulation length is often constrained by available computational resources. However, we have taken the reviewer's feedback seriously and have extended the simulation time to 300 ns to ensure adequate sampling of the peptide conformational changes and membrane interactions. Based on the analysis of root-mean-square deviation (RMSD) as well as other relevant parameters, we observed that the system reaches structural and conformational equilibrium after approximately 150 ns. We have carefully re-evaluated our analyses and updated the manuscript with the findings from this extended simulation.
- In general, the observed RMSD variations are not unexpected and are not suggestive of some large differences between the two peptides.
Response: We agree with the reviewer that the observed RMSD variations are not indicative of major structural differences between the two peptides. However, the peptides differ in their amino acid composition and net charge, which may influence their interaction dynamics with the membrane. Our intention is to explore how these differences affect peptide stability and membrane binding, and to further assess their binding affinities. This computational analysis is aimed at complementing our experimental findings and providing additional insights into their mechanism of action. The updated RMSD plot revealed significant differences between the two systems, and the information is incorporated in revised manuscript.
- The Rgs of the two peptides differ, because the authors' input models are different – one is fully folded into a helix, and the other one is rather disordered. Again, the chosen simulation time frame does not allow for deep conclusions based on this metric.
Response: We appreciate the reviewer’s observation. Indeed, the initial structural differences in the input models-one being predominantly helical and the other more disordered-do contribute to the variation observed in the radius of gyration (Rg). We acknowledge that the original simulation time frame was not sufficient to capture the complete conformational convergence. However, with the extended 300 ns simulations and based on the MD simulation movies, the peptides folding has significantly differed, and we have revised the relevant section in the manuscript to reflect this clarification and have interpreted the Rg data with appropriate caution.
- Lines 304–308: Comparing distributions between two independent PCAs of two different systems does not make any sense. You cannot extract any useful information from Fig. 6D. I suggest the authors try to understand what PCA is as a mathematical concept.
Response: Thank you for your insightful comment. We acknowledge that comparing principal component distributions from two independent PCA analyses of different systems is not methodologically appropriate, as it does not yield meaningful or interpretable information. We appreciate the reviewer’s clarification and have removed this comparison and the associated figure (Fig. 6D) from the revised manuscript accordingly.
- Section 3.4: So, these are MM-GBSA calculations from a single trajectory, is that right? You already have a rather limited sampling of the peptide—membrane interaction of just 100 ns. Relying on MM-GBSA calculations from a single trajectory can introduce significant artifacts, as it neglects conformational differences between the bound and unbound states; using separate trajectories for the complex, free peptide, and free membrane would better capture the distinct energetic contributions and would yield a more thermodynamically rigorous estimate of binding free energy.
Response: We thank the reviewer for highlighting the limitations of our MM-GBSA calculations from a single trajectory. Yes, the original MM-GBSA binding energy calculations were performed using a single trajectory approach based on 100 ns simulation data. We agree with the reviewer that this approach has limitations, especially given the short simulation time and the neglect of conformational differences between the bound and unbound states.
In response to additional suggestions, we have extended the simulations to 300 ns, and for the MM-GBSA calculations, we have now used snapshots from the most equilibrated portion of the trajectory (i.e., the last 100 ns, from 200 to 300 ns). While we acknowledge that using separate trajectories for the complex, free peptide, and free membrane would provide a more rigorous thermodynamic estimate, the single-trajectory approach is widely used and accepted in membrane-binding studies due to its computational efficiency and consistency in sampling.
Due to limited computational resources, multiple independent simulations for the unbound components were not feasible in our case. However, we believe that the updated analysis using the extended simulation time significantly improves the reliability of the binding energy estimates. The revised results have been included in the updated manuscript. We will prioritize performing these separate simulations in future work to further validate our findings.
- 7C: If GAGs on the bacterial envelope bind to GAGs on the host cells, wouldn’t it make sense to conduct in vitro experiments of when host cells are pretreated with G1, G2, or HS to check if they would competitively inhibit attachment of the EBs?
Response: As per the reviewer’s suggestions, we have now provided the data on when G1 and G2 peptides were pre-treated to the host HeLa and VK2 cells before challenging with Chlamydia. This data has been added as a supplementary figure 2 in the revised manuscript.
- Please, include a list of abbreviations for better readability.
Response: As per the reviewer’s suggestions we have now included the list of abbreviations on page in the revised manuscript.
Reviewer: 1
Comments to the Author
The manuscript presents an in vitro and in silico study of the effect of HS 3-O sulfation on the infectivity of Chlamydia muridarum. I find that the paper has many issues that require at least a major revision. In summary, the paper is not yet at a stage suitable for publication and would benefit from significant improvement.
This reviewer raised multiple good points which strengthened our findings. We agree with the reviewer’s comment and hence we have now provided substantial evidence by clarifying all the details throughout the manuscript.
Major revisions:
- I do not agree whatsoever with the authors' conclusions in the Abstract (lines 28–30) that their” results highlight the significance of 3-O sulfated heparan sulfate and other GAGs involved during the entry of muridarum into host cells.
Response: As per the reviewer’s recommendation, we have now carefully restated the conclusion in the abstract section that 3-O sulfated heparan sulfate recognizing G2 peptide prevents the entry of C. muridarum.
- Lines 57–60: These are proteoglycans and not GAGs. GAGs can be attached to core proteins to form PGs, but can also exist as free polymers, e.g. hyaluronic acid or heparin.
Response: As per the reviewer’s comment we have now corrected the statement on page of the revised manuscript.
- Some mention of the G1 and G2 peptides in the Introduction is necessary. What are these peptides? What evidence is there that they selectively bind HS and 3-OS HS specifically? Both peptides are highly positively charged, so I bet they would bind with high affinity to any heparin/HS chain, regardless of specific sulfation patterns.
Response: We have now discussed the origin and significance of G1/G2 peptides in the Introduction section in detail on page. We have also explained and cited our previous studies which have shown that removal of sulfated HS by heparinase III treatment results in significant loss of recognition of G2 peptide to cell surface sulfated HS (Ali et al., 2012 Journal of Virology).
We agree with the reviewer’s comments that due to the cationic nature of G1 and G2 peptides they may possibly interact with other heparin chains as well. Hence, we have now clearly stated that beside recognition of G2 peptide to HS/3-OS HS sites, the affinity of G2 peptide to other sites in HS chains or other pathogenic proteins can’t be ignored which may further be contributing towards the hindering the entry of Chlamydia muridarum. Since G1 and G2 peptides also appear to modulate HS expression (Park et al., 2013 IOVS) future studies will be helpful if the peptide exposure may influence Chlamydia–host cell interaction by gain or loss of HS expression. We have now discussed all the above possibilities in detail in the discussion section in the revised manuscript.
- Section 2.5.2: What was the specific composition of the Gram-negative membrane used? The authors need to provide detailed information on their computational model.
Response: Thank you very much for the suggestion. The lipid bilayer composition of the C. muridarum membrane, a Gram-negative bacterial pathogen, and the outer membrane of Chlamydia species contains a truncated LPS (long form) antigen with a group-specific epitope composed of a KDO trisaccharide (K-KDO-(2C8)-K-KDO-(2C4)-K-KDO). The inner leaflet of our model incorporated a mixture of PE, PG, and CL lipid headgroups in a 75:20:5 ratio. The information was obtained from the OPM database. We included these components in both our membrane systems. Moreover, these assembled peptide-membrane complexes were further used for MD simulations using Gromacs 2021.5 software. This information has been incorporated in the revised manuscript.
- Section 2.5.2: Was only the outer membrane used, or the whole bacterial envelope?
Response: In this study, the focus was primarily on the outer membrane, as Chlamydial outer membrane proteins (OMPs) play a crucial role in host cell interactions, primarily facilitating adhesion and entry.
- Section 2.5.2: I assume that since the CHARMM-GUI server was used for input generation, the force field for the MD simulations was CHARMM 36? Please, list your protocol in detail!
Response: We thank the reviewer for their thorough reading and the valuable suggestion to include a detailed protocol. Regarding the force field, we chose CHARMM36m due to its enhanced accuracy for simulating biomolecules, especially proteins and lipids, as it's a well-regarded refinement of the CHARMM36 all-atom additive force field. This information has been added to the revised manuscript for clarity.
- Lines 270–273 belong to the Materials and Methods section.
We have taken out the lines and moved it to material and method section.
- Lines 279–182: In general, 100 ns are highly insufficient to sample properly conformational changes in the peptides. In addition, when dealing with membranes, one expects hundreds or even thousands of ns of simulation time. I understand that simulation lengths depend on available computational resources. But to claim that 30 ns are sufficient for reaching both structural and conformational equilibrium is just ridiculous.
Response: We thank the reviewer for their critical assessment of the simulation length. We agree that 100 ns is a relatively short timescale for exploring the conformational landscape of peptides and their interactions with membranes, and we understand the preference for much longer simulations. As the reviewer correctly points out, simulation length is often constrained by available computational resources. However, we have taken the reviewer's feedback seriously and have extended the simulation time to 300 ns to ensure adequate sampling of the peptide conformational changes and membrane interactions. Based on the analysis of root-mean-square deviation (RMSD) as well as other relevant parameters, we observed that the system reaches structural and conformational equilibrium after approximately 150 ns. We have carefully re-evaluated our analyses and updated the manuscript with the findings from this extended simulation.
- In general, the observed RMSD variations are not unexpected and are not suggestive of some large differences between the two peptides.
Response: We agree with the reviewer that the observed RMSD variations are not indicative of major structural differences between the two peptides. However, the peptides differ in their amino acid composition and net charge, which may influence their interaction dynamics with the membrane. Our intention is to explore how these differences affect peptide stability and membrane binding, and to further assess their binding affinities. This computational analysis is aimed at complementing our experimental findings and providing additional insights into their mechanism of action. The updated RMSD plot revealed significant differences between the two systems, and the information is incorporated in revised manuscript.
- The Rgs of the two peptides differ, because the authors' input models are different – one is fully folded into a helix, and the other one is rather disordered. Again, the chosen simulation time frame does not allow for deep conclusions based on this metric.
Response: We appreciate the reviewer’s observation. Indeed, the initial structural differences in the input models-one being predominantly helical and the other more disordered-do contribute to the variation observed in the radius of gyration (Rg). We acknowledge that the original simulation time frame was not sufficient to capture the complete conformational convergence. However, with the extended 300 ns simulations and based on the MD simulation movies, the peptides folding has significantly differed, and we have revised the relevant section in the manuscript to reflect this clarification and have interpreted the Rg data with appropriate caution.
- Lines 304–308: Comparing distributions between two independent PCAs of two different systems does not make any sense. You cannot extract any useful information from Fig. 6D. I suggest the authors try to understand what PCA is as a mathematical concept.
Response: Thank you for your insightful comment. We acknowledge that comparing principal component distributions from two independent PCA analyses of different systems is not methodologically appropriate, as it does not yield meaningful or interpretable information. We appreciate the reviewer’s clarification and have removed this comparison and the associated figure (Fig. 6D) from the revised manuscript accordingly.
- Section 3.4: So, these are MM-GBSA calculations from a single trajectory, is that right? You already have a rather limited sampling of the peptide—membrane interaction of just 100 ns. Relying on MM-GBSA calculations from a single trajectory can introduce significant artifacts, as it neglects conformational differences between the bound and unbound states; using separate trajectories for the complex, free peptide, and free membrane would better capture the distinct energetic contributions and would yield a more thermodynamically rigorous estimate of binding free energy.
Response: We thank the reviewer for highlighting the limitations of our MM-GBSA calculations from a single trajectory. Yes, the original MM-GBSA binding energy calculations were performed using a single trajectory approach based on 100 ns simulation data. We agree with the reviewer that this approach has limitations, especially given the short simulation time and the neglect of conformational differences between the bound and unbound states.
In response to additional suggestions, we have extended the simulations to 300 ns, and for the MM-GBSA calculations, we have now used snapshots from the most equilibrated portion of the trajectory (i.e., the last 100 ns, from 200 to 300 ns). While we acknowledge that using separate trajectories for the complex, free peptide, and free membrane would provide a more rigorous thermodynamic estimate, the single-trajectory approach is widely used and accepted in membrane-binding studies due to its computational efficiency and consistency in sampling.
Due to limited computational resources, multiple independent simulations for the unbound components were not feasible in our case. However, we believe that the updated analysis using the extended simulation time significantly improves the reliability of the binding energy estimates. The revised results have been included in the updated manuscript. We will prioritize performing these separate simulations in future work to further validate our findings.
- 7C: If GAGs on the bacterial envelope bind to GAGs on the host cells, wouldn’t it make sense to conduct in vitro experiments of when host cells are pretreated with G1, G2, or HS to check if they would competitively inhibit attachment of the EBs?
Response: As per the reviewer’s suggestions, we have now provided the data on when G1 and G2 peptides were pre-treated to the host HeLa and VK2 cells before challenging with Chlamydia. This data has been added as a supplementary figure 2 in the revised manuscript.
- Please, include a list of abbreviations for better readability.
Response: As per the reviewer’s suggestions we have now included the list of abbreviations on page in the revised manuscript.
Reviewer: 1
Comments to the Author
The manuscript presents an in vitro and in silico study of the effect of HS 3-O sulfation on the infectivity of Chlamydia muridarum. I find that the paper has many issues that require at least a major revision. In summary, the paper is not yet at a stage suitable for publication and would benefit from significant improvement.
This reviewer raised multiple good points which strengthened our findings. We agree with the reviewer’s comment and hence we have now provided substantial evidence by clarifying all the details throughout the manuscript.
Major revisions:
- I do not agree whatsoever with the authors' conclusions in the Abstract (lines 28–30) that their” results highlight the significance of 3-O sulfated heparan sulfate and other GAGs involved during the entry of muridarum into host cells.
Response: As per the reviewer’s recommendation, we have now carefully restated the conclusion in the abstract section that 3-O sulfated heparan sulfate recognizing G2 peptide prevents the entry of C. muridarum.
- Lines 57–60: These are proteoglycans and not GAGs. GAGs can be attached to core proteins to form PGs, but can also exist as free polymers, e.g. hyaluronic acid or heparin.
Response: As per the reviewer’s comment we have now corrected the statement on page of the revised manuscript.
- Some mention of the G1 and G2 peptides in the Introduction is necessary. What are these peptides? What evidence is there that they selectively bind HS and 3-OS HS specifically? Both peptides are highly positively charged, so I bet they would bind with high affinity to any heparin/HS chain, regardless of specific sulfation patterns.
Response: We have now discussed the origin and significance of G1/G2 peptides in the Introduction section in detail on page. We have also explained and cited our previous studies which have shown that removal of sulfated HS by heparinase III treatment results in significant loss of recognition of G2 peptide to cell surface sulfated HS (Ali et al., 2012 Journal of Virology).
We agree with the reviewer’s comments that due to the cationic nature of G1 and G2 peptides they may possibly interact with other heparin chains as well. Hence, we have now clearly stated that beside recognition of G2 peptide to HS/3-OS HS sites, the affinity of G2 peptide to other sites in HS chains or other pathogenic proteins can’t be ignored which may further be contributing towards the hindering the entry of Chlamydia muridarum. Since G1 and G2 peptides also appear to modulate HS expression (Park et al., 2013 IOVS) future studies will be helpful if the peptide exposure may influence Chlamydia–host cell interaction by gain or loss of HS expression. We have now discussed all the above possibilities in detail in the discussion section in the revised manuscript.
- Section 2.5.2: What was the specific composition of the Gram-negative membrane used? The authors need to provide detailed information on their computational model.
Response: Thank you very much for the suggestion. The lipid bilayer composition of the C. muridarum membrane, a Gram-negative bacterial pathogen, and the outer membrane of Chlamydia species contains a truncated LPS (long form) antigen with a group-specific epitope composed of a KDO trisaccharide (K-KDO-(2C8)-K-KDO-(2C4)-K-KDO). The inner leaflet of our model incorporated a mixture of PE, PG, and CL lipid headgroups in a 75:20:5 ratio. The information was obtained from the OPM database. We included these components in both our membrane systems. Moreover, these assembled peptide-membrane complexes were further used for MD simulations using Gromacs 2021.5 software. This information has been incorporated in the revised manuscript.
- Section 2.5.2: Was only the outer membrane used, or the whole bacterial envelope?
Response: In this study, the focus was primarily on the outer membrane, as Chlamydial outer membrane proteins (OMPs) play a crucial role in host cell interactions, primarily facilitating adhesion and entry.
- Section 2.5.2: I assume that since the CHARMM-GUI server was used for input generation, the force field for the MD simulations was CHARMM 36? Please, list your protocol in detail!
Response: We thank the reviewer for their thorough reading and the valuable suggestion to include a detailed protocol. Regarding the force field, we chose CHARMM36m due to its enhanced accuracy for simulating biomolecules, especially proteins and lipids, as it's a well-regarded refinement of the CHARMM36 all-atom additive force field. This information has been added to the revised manuscript for clarity.
- Lines 270–273 belong to the Materials and Methods section.
We have taken out the lines and moved it to material and method section.
- Lines 279–182: In general, 100 ns are highly insufficient to sample properly conformational changes in the peptides. In addition, when dealing with membranes, one expects hundreds or even thousands of ns of simulation time. I understand that simulation lengths depend on available computational resources. But to claim that 30 ns are sufficient for reaching both structural and conformational equilibrium is just ridiculous.
Response: We thank the reviewer for their critical assessment of the simulation length. We agree that 100 ns is a relatively short timescale for exploring the conformational landscape of peptides and their interactions with membranes, and we understand the preference for much longer simulations. As the reviewer correctly points out, simulation length is often constrained by available computational resources. However, we have taken the reviewer's feedback seriously and have extended the simulation time to 300 ns to ensure adequate sampling of the peptide conformational changes and membrane interactions. Based on the analysis of root-mean-square deviation (RMSD) as well as other relevant parameters, we observed that the system reaches structural and conformational equilibrium after approximately 150 ns. We have carefully re-evaluated our analyses and updated the manuscript with the findings from this extended simulation.
- In general, the observed RMSD variations are not unexpected and are not suggestive of some large differences between the two peptides.
Response: We agree with the reviewer that the observed RMSD variations are not indicative of major structural differences between the two peptides. However, the peptides differ in their amino acid composition and net charge, which may influence their interaction dynamics with the membrane. Our intention is to explore how these differences affect peptide stability and membrane binding, and to further assess their binding affinities. This computational analysis is aimed at complementing our experimental findings and providing additional insights into their mechanism of action. The updated RMSD plot revealed significant differences between the two systems, and the information is incorporated in revised manuscript.
- The Rgs of the two peptides differ, because the authors' input models are different – one is fully folded into a helix, and the other one is rather disordered. Again, the chosen simulation time frame does not allow for deep conclusions based on this metric.
Response: We appreciate the reviewer’s observation. Indeed, the initial structural differences in the input models-one being predominantly helical and the other more disordered-do contribute to the variation observed in the radius of gyration (Rg). We acknowledge that the original simulation time frame was not sufficient to capture the complete conformational convergence. However, with the extended 300 ns simulations and based on the MD simulation movies, the peptides folding has significantly differed, and we have revised the relevant section in the manuscript to reflect this clarification and have interpreted the Rg data with appropriate caution.
- Lines 304–308: Comparing distributions between two independent PCAs of two different systems does not make any sense. You cannot extract any useful information from Fig. 6D. I suggest the authors try to understand what PCA is as a mathematical concept.
Response: Thank you for your insightful comment. We acknowledge that comparing principal component distributions from two independent PCA analyses of different systems is not methodologically appropriate, as it does not yield meaningful or interpretable information. We appreciate the reviewer’s clarification and have removed this comparison and the associated figure (Fig. 6D) from the revised manuscript accordingly.
- Section 3.4: So, these are MM-GBSA calculations from a single trajectory, is that right? You already have a rather limited sampling of the peptide—membrane interaction of just 100 ns. Relying on MM-GBSA calculations from a single trajectory can introduce significant artifacts, as it neglects conformational differences between the bound and unbound states; using separate trajectories for the complex, free peptide, and free membrane would better capture the distinct energetic contributions and would yield a more thermodynamically rigorous estimate of binding free energy.
Response: We thank the reviewer for highlighting the limitations of our MM-GBSA calculations from a single trajectory. Yes, the original MM-GBSA binding energy calculations were performed using a single trajectory approach based on 100 ns simulation data. We agree with the reviewer that this approach has limitations, especially given the short simulation time and the neglect of conformational differences between the bound and unbound states.
In response to additional suggestions, we have extended the simulations to 300 ns, and for the MM-GBSA calculations, we have now used snapshots from the most equilibrated portion of the trajectory (i.e., the last 100 ns, from 200 to 300 ns). While we acknowledge that using separate trajectories for the complex, free peptide, and free membrane would provide a more rigorous thermodynamic estimate, the single-trajectory approach is widely used and accepted in membrane-binding studies due to its computational efficiency and consistency in sampling.
Due to limited computational resources, multiple independent simulations for the unbound components were not feasible in our case. However, we believe that the updated analysis using the extended simulation time significantly improves the reliability of the binding energy estimates. The revised results have been included in the updated manuscript. We will prioritize performing these separate simulations in future work to further validate our findings.
- 7C: If GAGs on the bacterial envelope bind to GAGs on the host cells, wouldn’t it make sense to conduct in vitro experiments of when host cells are pretreated with G1, G2, or HS to check if they would competitively inhibit attachment of the EBs?
Response: As per the reviewer’s suggestions, we have now provided the data on when G1 and G2 peptides were pre-treated to the host HeLa and VK2 cells before challenging with Chlamydia. This data has been added as a supplementary figure 2 in the revised manuscript.
- Please, include a list of abbreviations for better readability.
Response: As per the reviewer’s suggestions we have now included the list of abbreviations on page in the revised manuscript.
Reviewer: 1
Comments to the Author
The manuscript presents an in vitro and in silico study of the effect of HS 3-O sulfation on the infectivity of Chlamydia muridarum. I find that the paper has many issues that require at least a major revision. In summary, the paper is not yet at a stage suitable for publication and would benefit from significant improvement.
This reviewer raised multiple good points which strengthened our findings. We agree with the reviewer’s comment and hence we have now provided substantial evidence by clarifying all the details throughout the manuscript.
Major revisions:
- I do not agree whatsoever with the authors' conclusions in the Abstract (lines 28–30) that their” results highlight the significance of 3-O sulfated heparan sulfate and other GAGs involved during the entry of muridarum into host cells.
Response: As per the reviewer’s recommendation, we have now carefully restated the conclusion in the abstract section that 3-O sulfated heparan sulfate recognizing G2 peptide prevents the entry of C. muridarum.
- Lines 57–60: These are proteoglycans and not GAGs. GAGs can be attached to core proteins to form PGs, but can also exist as free polymers, e.g. hyaluronic acid or heparin.
Response: As per the reviewer’s comment we have now corrected the statement on page of the revised manuscript.
- Some mention of the G1 and G2 peptides in the Introduction is necessary. What are these peptides? What evidence is there that they selectively bind HS and 3-OS HS specifically? Both peptides are highly positively charged, so I bet they would bind with high affinity to any heparin/HS chain, regardless of specific sulfation patterns.
Response: We have now discussed the origin and significance of G1/G2 peptides in the Introduction section in detail on page. We have also explained and cited our previous studies which have shown that removal of sulfated HS by heparinase III treatment results in significant loss of recognition of G2 peptide to cell surface sulfated HS (Ali et al., 2012 Journal of Virology).
We agree with the reviewer’s comments that due to the cationic nature of G1 and G2 peptides they may possibly interact with other heparin chains as well. Hence, we have now clearly stated that beside recognition of G2 peptide to HS/3-OS HS sites, the affinity of G2 peptide to other sites in HS chains or other pathogenic proteins can’t be ignored which may further be contributing towards the hindering the entry of Chlamydia muridarum. Since G1 and G2 peptides also appear to modulate HS expression (Park et al., 2013 IOVS) future studies will be helpful if the peptide exposure may influence Chlamydia–host cell interaction by gain or loss of HS expression. We have now discussed all the above possibilities in detail in the discussion section in the revised manuscript.
- Section 2.5.2: What was the specific composition of the Gram-negative membrane used? The authors need to provide detailed information on their computational model.
Response: Thank you very much for the suggestion. The lipid bilayer composition of the C. muridarum membrane, a Gram-negative bacterial pathogen, and the outer membrane of Chlamydia species contains a truncated LPS (long form) antigen with a group-specific epitope composed of a KDO trisaccharide (K-KDO-(2C8)-K-KDO-(2C4)-K-KDO). The inner leaflet of our model incorporated a mixture of PE, PG, and CL lipid headgroups in a 75:20:5 ratio. The information was obtained from the OPM database. We included these components in both our membrane systems. Moreover, these assembled peptide-membrane complexes were further used for MD simulations using Gromacs 2021.5 software. This information has been incorporated in the revised manuscript.
- Section 2.5.2: Was only the outer membrane used, or the whole bacterial envelope?
Response: In this study, the focus was primarily on the outer membrane, as Chlamydial outer membrane proteins (OMPs) play a crucial role in host cell interactions, primarily facilitating adhesion and entry.
- Section 2.5.2: I assume that since the CHARMM-GUI server was used for input generation, the force field for the MD simulations was CHARMM 36? Please, list your protocol in detail!
Response: We thank the reviewer for their thorough reading and the valuable suggestion to include a detailed protocol. Regarding the force field, we chose CHARMM36m due to its enhanced accuracy for simulating biomolecules, especially proteins and lipids, as it's a well-regarded refinement of the CHARMM36 all-atom additive force field. This information has been added to the revised manuscript for clarity.
- Lines 270–273 belong to the Materials and Methods section.
We have taken out the lines and moved it to material and method section.
- Lines 279–182: In general, 100 ns are highly insufficient to sample properly conformational changes in the peptides. In addition, when dealing with membranes, one expects hundreds or even thousands of ns of simulation time. I understand that simulation lengths depend on available computational resources. But to claim that 30 ns are sufficient for reaching both structural and conformational equilibrium is just ridiculous.
Response: We thank the reviewer for their critical assessment of the simulation length. We agree that 100 ns is a relatively short timescale for exploring the conformational landscape of peptides and their interactions with membranes, and we understand the preference for much longer simulations. As the reviewer correctly points out, simulation length is often constrained by available computational resources. However, we have taken the reviewer's feedback seriously and have extended the simulation time to 300 ns to ensure adequate sampling of the peptide conformational changes and membrane interactions. Based on the analysis of root-mean-square deviation (RMSD) as well as other relevant parameters, we observed that the system reaches structural and conformational equilibrium after approximately 150 ns. We have carefully re-evaluated our analyses and updated the manuscript with the findings from this extended simulation.
- In general, the observed RMSD variations are not unexpected and are not suggestive of some large differences between the two peptides.
Response: We agree with the reviewer that the observed RMSD variations are not indicative of major structural differences between the two peptides. However, the peptides differ in their amino acid composition and net charge, which may influence their interaction dynamics with the membrane. Our intention is to explore how these differences affect peptide stability and membrane binding, and to further assess their binding affinities. This computational analysis is aimed at complementing our experimental findings and providing additional insights into their mechanism of action. The updated RMSD plot revealed significant differences between the two systems, and the information is incorporated in revised manuscript.
- The Rgs of the two peptides differ, because the authors' input models are different – one is fully folded into a helix, and the other one is rather disordered. Again, the chosen simulation time frame does not allow for deep conclusions based on this metric.
Response: We appreciate the reviewer’s observation. Indeed, the initial structural differences in the input models-one being predominantly helical and the other more disordered-do contribute to the variation observed in the radius of gyration (Rg). We acknowledge that the original simulation time frame was not sufficient to capture the complete conformational convergence. However, with the extended 300 ns simulations and based on the MD simulation movies, the peptides folding has significantly differed, and we have revised the relevant section in the manuscript to reflect this clarification and have interpreted the Rg data with appropriate caution.
- Lines 304–308: Comparing distributions between two independent PCAs of two different systems does not make any sense. You cannot extract any useful information from Fig. 6D. I suggest the authors try to understand what PCA is as a mathematical concept.
Response: Thank you for your insightful comment. We acknowledge that comparing principal component distributions from two independent PCA analyses of different systems is not methodologically appropriate, as it does not yield meaningful or interpretable information. We appreciate the reviewer’s clarification and have removed this comparison and the associated figure (Fig. 6D) from the revised manuscript accordingly.
- Section 3.4: So, these are MM-GBSA calculations from a single trajectory, is that right? You already have a rather limited sampling of the peptide—membrane interaction of just 100 ns. Relying on MM-GBSA calculations from a single trajectory can introduce significant artifacts, as it neglects conformational differences between the bound and unbound states; using separate trajectories for the complex, free peptide, and free membrane would better capture the distinct energetic contributions and would yield a more thermodynamically rigorous estimate of binding free energy.
Response: We thank the reviewer for highlighting the limitations of our MM-GBSA calculations from a single trajectory. Yes, the original MM-GBSA binding energy calculations were performed using a single trajectory approach based on 100 ns simulation data. We agree with the reviewer that this approach has limitations, especially given the short simulation time and the neglect of conformational differences between the bound and unbound states.
In response to additional suggestions, we have extended the simulations to 300 ns, and for the MM-GBSA calculations, we have now used snapshots from the most equilibrated portion of the trajectory (i.e., the last 100 ns, from 200 to 300 ns). While we acknowledge that using separate trajectories for the complex, free peptide, and free membrane would provide a more rigorous thermodynamic estimate, the single-trajectory approach is widely used and accepted in membrane-binding studies due to its computational efficiency and consistency in sampling.
Due to limited computational resources, multiple independent simulations for the unbound components were not feasible in our case. However, we believe that the updated analysis using the extended simulation time significantly improves the reliability of the binding energy estimates. The revised results have been included in the updated manuscript. We will prioritize performing these separate simulations in future work to further validate our findings.
- 7C: If GAGs on the bacterial envelope bind to GAGs on the host cells, wouldn’t it make sense to conduct in vitro experiments of when host cells are pretreated with G1, G2, or HS to check if they would competitively inhibit attachment of the EBs?
Response: As per the reviewer’s suggestions, we have now provided the data on when G1 and G2 peptides were pre-treated to the host HeLa and VK2 cells before challenging with Chlamydia. This data has been added as a supplementary figure 2 in the revised manuscript.
- Please, include a list of abbreviations for better readability.
Response: As per the reviewer’s suggestions we have now included the list of abbreviations on page in the revised manuscript.
Reviewer 2 Report
Comments and Suggestions for Authors
I think this is a well-executed study with significant implications for understanding Chlamydia pathogenesis. Some problems should be solved to make it better.
-
The results in Figures 3 and 4 are intriguing, but the presentation could be more intuitive. For instance, the differences in G1 and G2 peptide effects between HeLa and VK2 cells would be clearer if the figure legends explicitly highlighted the contrasting outcomes. A side-by-side comparison of the two cell lines might better emphasize the cell-specific responses. Additionally, labeling the exact peptide concentrations in the figure panels would help readers quickly grasp the dose-dependent effects.
-
The finding that G1 enhances infectivity in HeLa cells but inhibits it in VK2 cells is surprising and warrants deeper discussion. Is this due to differences in HS sulfation patterns between the cell types, or could it reflect distinct receptor usage by C. muridarum in epithelial vs. cervical cells? The proposed model in Figure 7 is compelling, but some experimental validation—such as competitive binding assays with purified HS variants—would strengthen the argument that G2 directly targets sulfated moieties on the bacterial surface.
-
The statistical analysis appears sound, but including exact *p*-values (rather than just *p* < 0.05) would provide a more transparent assessment of significance. The Methods section should also clarify the number of biological replicates, as this is critical for evaluating reproducibility. Were experiments repeated independently, or were the replicates technical (e.g., multiple wells from the same cell passage)? This distinction matters for interpreting variability.
-
The discussion nicely contrasted Chlamydia and HSV-1 entry mechanisms, but it could also address how these findings relate to other HS-dependent pathogens (e.g., Toxoplasma or RSV). Do these microbes similarly exploit 3-O sulfation, or is this unique to Chlamydia? The conclusion raises an important point about potential serovar-specific effects of G2—could the authors speculate on whether this peptide might also neutralize clinical C. trachomatis strains, or would modifications be needed?
Author Response
Reviewer: 2
- The results in Figures 3 and 4 are intriguing, but the presentation could be more intuitive. For instance, the differences in G1 and G2 peptide effects between HeLa and VK2 cells would be clearer if the figure legends explicitly highlighted the contrasting outcomes. A side-by-side comparison of the two cell lines might better emphasize the cell-specific responses. Additionally, labeling the exact peptide concentrations in the figure panels would help readers quickly grasp the dose-dependent effects.
Response: As per reviewer’s suggestions we have clarified the cell-dependent effect of G2 peptide in Figure legends. In addition, we have now labeled the exact peptide concentrations in the figure panels.
- The finding that G1 enhances infectivity in HeLa cells but inhibits it in VK2 cells is surprising and warrants deeper discussion. Is this due to differences in HS sulfation patterns between the cell types, or could it reflect distinct receptor usage by muridarumin epithelial vs. cervical cells? The proposed model in Figure 7 is compelling, but some experimental validations such as competitive binding assays with purified HS variants—would strengthen the argument that G2 directly targets sulfated moieties on the bacterial surface.
This is a very interesting point raised by the reviewer. Accordingly, we have discussed the possibility of differences in the HS and or 3-OS HS patterns. In fact, G1 and G2 peptides are known modulators of HS expression (Suryawanshi et al., 2021 Pathogens; Park et al., 2013 IOVS), therefore we have discussed the potential possibilities in the terms of HS and sulfated HS expression in HeLa and VK2 cells which may have impacted the variability in our results to the peptide exposure. We have also added to our previous finding that G2 peptide recognizes sulfated HS (Ali et al., 2012 Journal of Virology).
- The statistical analysis appears sound, but including exact *p*-values (rather than just *p* < 0.05) would provide a more transparent assessment of significance. The Methods section should also clarify the number of biological replicates, as this is critical for evaluating reproducibility. Were experiments repeated independently, or were the replicates technical (e.g., multiple wells from the same cell passage)? This distinction matters for interpreting variability.
Response: As per the suggestions, we have now used exact p-values in the graphs. In addition, each figure legend has clarified regarding three independent biological replicates used in this study.
- The discussion nicely contrasted with Chlamydiaand HSV-1 entry mechanisms, but it could also address how these findings relate to other HS-dependent pathogens (e.g., Toxoplasma or RSV). Do these microbes similarly exploit 3-O sulfation, or is this unique to Chlamydia? The conclusion raises an important point about potential serovar-specific effects of G2—could the authors speculate on whether this peptide might also neutralize clinical trachomatis strains, or would modifications be needed?
Response: As per the reviewer’s suggestion, we have now provided detailed comparison to our current findings with other pathogens in terms of HS-dependent usage. As pointed out by the reviewer we have also discussed the possibility of testing the impact of G2 peptide on human strains of Chlamydia trachomatis and the associated serovars as they utilize a diverse subset of sulfated HS to promote cell entry (Fechtner et al., 2013 Journal of Bacteriology).

Reviewer 3 Report
Comments and Suggestions for Authors
The present manuscript "ID:biomolecules-3631125" titled "3-O sulfated heparan sulfate (G2) peptide ligand impairs infectivity of Chlamydia muridarum" shows that 3-O-sulfated HS), can significantly block the infectivity of Chlamydia muridarum by binding directly to structures on the bacterial surface. It also demonstrates that host cell expression of 3-O-sulfated HS is not essential for the bacteria to enter cells, suggesting that the peptide’s action is likely on the bacterium itself rather than on the host cell. This finding opens the door to new antimicrobial strategies targeting bacterial surface components. However, there are many weaknesses associated with the manuscript that should be addressed.
Abstract. The abstract clearly states the purpose of the study and the key finding that the G2 peptide reduces Chlamydia infectivity. The authors need to provide important quantitative data in this section.
Introduction. It provides a solid background on Chlamydia infections and why studying heparan sulfate is important.
Materials and Methods. While multiple concentrations of peptides are tested, the design does not include time-course experiments or long-term infection models, which would give a more complete picture of how the peptides affect infection dynamics.
Results. The results are organized in a logical order and backed up with quantitative data, showing clear effects of the G2 peptide. However, the figures (e.g., Fig. 2–6) are not explained well in text, visual data is not well-integrated with the writing. Moreover, some comparisons (like between G1 and G2 effects) are not fully interpreted, which weakens the clarity of findings.
Discussion. It repeats earlier information instead of offering new insights or clearly discussing limitations. The conclusion about the G2 peptide’s potential use as therapy is not supported by in vivo or long-term studies.
More comments:
- For some complex methods (like the simulation pipeline), a flowchart or schematic would make it much easier to follow the steps and understand what’s being modeled.
- The therapeutic angle and generalization to broader contexts need more evidence.
Author Response
Reviewer 3
The present manuscript "ID: biomolecules-3631125" titled "3-O sulfated heparan sulfate (G2) peptide ligand impairs infectivity of Chlamydia muridarum" shows that 3-O-sulfated HS), can significantly block the infectivity of Chlamydia muridarum by binding directly to structures on the bacterial surface. It also demonstrates that host cell expression of 3-O-sulfated HS is not essential for the bacteria to enter cells, suggesting that the peptide’s action is likely on the bacterium itself rather than on the host cell. This finding opens the door to new antimicrobial strategies targeting bacterial surface components. However, there are many weaknesses associated with the manuscript that should be addressed.
Abstract. The abstract clearly states the purpose of the study and the key findings that the G2 peptide reduces Chlamydia infectivity. The authors need to provide important quantitative data in this section.
Response: The data shown in Fig. 3 and Fig. 4 highlights the effect of G2 peptides in reducing the number of inclusion bodies formed by EBs of C. muridarum in both HeLa and VK2 cells. A similar but less pronounced effect was observed when the host cells were pretreated with the peptides before the infection. This new data has been added to the supplementary figure 2.
Introduction. It provides a solid background on Chlamydia infections and why studying heparan sulfate is important.
Materials and Methods. While multiple concentrations of peptides are tested, the design does not include time-course experiments or long-term infection models, which would give a more complete picture of how the peptides affect infection dynamics.
Response: We used early time points only since our aim was to detect the effect of peptides on Chlamydia cell entry which normally takes a few hours. We agree with the reviewer to study the impact of peptides on infection dynamics which we plan to study in the next phase of our research project.
Results. The results are organized in a logical order and backed up with quantitative data, showing clear effects of the G2 peptide. However, the figures (e.g., Fig. 2–6) are not explained well in text, visual data is not well-integrated with the writing. Moreover, some comparisons (like between G1 and G2 effects) are not fully interpreted, which weakens the clarity of findings.
Response: As per the recommendations of the reviewer we have explained Fig. 2-to-Fig. 6 well enough in the text including integrating the visual data into the text. We have also compared the results of G1 and G2 peptides.
Discussion. It repeats earlier information instead of offering new insights or clearly discussing limitations. The conclusion about the G2 peptide’s potential use as therapy is not supported by in vivo or long-term studies.
Response: Due to the lack of enough funds, we were not able to perform testing our in vitro findings in the vaginal murine model of Chlamydia. In future we are determined to continue our research by utilizing vivo studies when the funds are available. However, we have significantly improved our discussion section with new insights into future goals including covering the limitations in this study.
More comments:
- For some complex methods (like the simulation pipeline), a flowchart or schematic would make it much easier to follow the steps and understand what’s being modeled.
Response: Thank you for this useful suggestion. As per your advice we have added a flowchart for our simulation studies as Supplementary Figure 3 in the revised manuscript.
- The therapeutic angle and generalization to broader contexts need more evidence.
Response: We agree with the reviewer’s comment. Upon the availability of additional funds, we are determined to carry out the next step of testing by using multiple other serovars including HS binding assays.

Round 2
Reviewer 3 Report
Comments and Suggestions for Authors
The authors have responded to my comments in the revised manuscript.